# TAZ-CAMTA1 and YAP-TFE3 alter the TAZ/YAP transcriptome by recruiting the ATAC histone acetyltransferase complex

**Nicole Merritt[1†], Keith Garcia[1,2†], Dushyandi Rajendran[3], Zhen-Yuan Lin[3], Xiaomeng Zhang[4], Katrina A Mitchell[4,5], Nicholas Borcherding[6], Colleen Fullenkamp[1], Michael S Chimenti[7], Anne-Claude Gingras[3], Kieran F Harvey[4,5,8], Munir R Tanas[1,2,9,10]***

[1]Department of Pathology, University of Iowa, Iowa City, United States; [2]Cancer Biology Graduate Program, University of Iowa, Iowa City, United States; [3]Lunenfeld-Tanenbaum Research Institute, Mount Sinai Hospital, Toronto, United States; [4]Peter MacCallum Cancer Centre, Melbourne, Australia; [5]Sir Peter MacCallum Department of Oncology, The University of Melbourne, Parkville, Australia; [6]Department of Pathology and Immunology, Washington University, St. Louis, United States; [7]Iowa Institute of Human Genetics, Carver College of Medicine, University of Iowa, Iowa City, United States; [8]Department of Anatomy and Developmental Biology and Biomedicine Discovery Institute, Monash University, Clayton, Australia; [9]Holden Comprehensive Cancer Center, University of Iowa, Iowa City, United States; [10]Pathology and Laboratory Medicine, Veterans Affairs Medical Center, Iowa City, United States

*For correspondence: munir-tanas@uiowa.edu

†These authors contributed equally to this work

**Competing interests:** The authors declare that no competing interests exist.

**Abstract** Epithelioid hemangioendothelioma (EHE) is a vascular sarcoma that metastasizes early in its clinical course and lacks an effective medical therapy. The TAZ-CAMTA1 and YAP-TFE3 fusion proteins are chimeric transcription factors and initiating oncogenic drivers of EHE. A combined proteomic/genetic screen in human cell lines identified YEATS2 and ZZZ3, components of the Ada2a-containing histone acetyltransferase (ATAC) complex, as key interactors of both fusion proteins despite the dissimilarity of the C terminal fusion partners CAMTA1 and TFE3. Integrative next-generation sequencing approaches in human and murine cell lines showed that the fusion proteins drive a unique transcriptome by simultaneously hyperactivating a TEAD-based transcriptional program and modulating the chromatin environment via interaction with the ATAC complex. Interaction of the ATAC complex with both fusion proteins indicates that it is a key oncogenic driver and unifying enzymatic therapeutic target for this sarcoma. This study presents an approach to mechanistically dissect how chimeric transcription factors drive the formation of human cancers.

## Introduction

Epithelioid hemangioendothelioma (EHE) is a vascular sarcoma that arises over a wide age range and can occur essentially anywhere anatomically. When it occurs in the lung or the liver, it frequently presents with metastasis early in its clinical course (*Goldblum and Weiss, 2014*). Two mutually exclusive gene fusions define this sarcoma which lacks an effective medical therapy. The *WWTR1-CAMTA1* gene fusion is encoded by a t(1;3)(p36;q25) chromosomal translocation (*Errani et al., 2011*, *Tanas et al., 2011*; *Tanas et al., 2016*) resulting in a TAZ-CAMTA1 fusion protein present in 85–90% of all EHE tumors (*Errani et al., 2011*; *Tanas et al., 2011*). The other ~10–15% of EHE

**eLife digest** The proliferation of human cells is tightly regulated to ensure that enough cells are made to build and repair organs and tissues, while at the same time stopping cells from dividing uncontrollably and damaging the body. To get the right balance, cells rely on physical and chemical cues from their environment that trigger the biochemical signals that regulate two proteins called TAZ and YAP. These proteins control gene activity by regulating the rate at which genes are copied to produce proteins. If this process becomes dysregulated, cells can grow uncontrollably, causing cancer.

In cancer cells, it is common to find TAZ and YAP fused to other proteins. In epithelioid hemangioendothelioma, a rare cancer that grows in the blood vessels, cancerous growth can be driven by a version of TAZ fused to the protein CAMTA1, or a version of YAP fused to the protein TFE3. While the role of TAZ and YAP in promoting gene activity is known, it is unclear how CAMTA1 and TFE3 contribute to cell growth becoming dysregulated.

Merritt, Garcia et al. studied sarcoma cell lines to show that these two fusion proteins, TAZ-CAMTA1 and YAP-TFE3, change the pattern of gene activity seen in the cells compared to TAZ or YAP alone. An analysis of molecules that interact with the two fusion proteins identified a complex called ATAC as the cause of these changes. This complex adds chemical markers to DNA-packaging proteins, which control which genes are available for activation. The fusion proteins combine the ability of TAZ and YAP to control gene activity and the ability of CAMTA1 and TFE3 to make DNA more accessible, allowing the fusion proteins to drive uncontrolled cancerous growth.

Similar TAZ and YAP fusion proteins have been found in other cancers, which can activate genes and potentially alter DNA packaging. Targeting drug development efforts at the proteins that complex with TAZ and YAP fusion proteins may lead to new therapies.

contain a t(X;11) (p11;q22) chromosomal translocation encoding a *YAP1-TFE3* gene fusion (*Antonescu et al., 2013*). Homologous to TAZ-CAMTA1, YAP-TFE3 fuses the N-terminus of YAP in frame to the C-terminus of TFE3 (transcription factor E3) (*Antonescu et al., 2013*).

The Hippo pathway is a serine/threonine kinase cascade involved in regulating tissue growth and organ size and is composed of the STE20-like protein kinases 1 and 2 (MST1/2; homologs of *Drosophila* Hpo) (*Harvey et al., 2003*; *Wu et al., 2003*; *Udan et al., 2003*; *Pantalacci et al., 2003*; *Dong et al., 2007*; *Zhao et al., 2007*) and the large tumor suppressor 1 and 2 (LATS1/2; homologs of *Drosophila* Wts) (*Justice et al., 1995*; *Xu et al., 1995*; *Tapon et al., 2002*). In mammalian cells, the nuclear effectors of the Hippo pathway are TAZ (gene is *WWTR1*) and YAP, two paralogous transcriptional coactivators (*Dong et al., 2007*; *Zhao et al., 2007*; *Camargo et al., 2007*). Hippo signaling limits TAZ and YAP activity by phosphorylating various serine residues, promoting their accumulation in the cytoplasm and ubiquitin-mediated degradation (*Dong et al., 2007*; *Zhao et al., 2007*). In the nucleus, TAZ and YAP bind to the TEA domain (TEAD) family of transcription factors to drive their transcriptional program (*Zhao et al., 2008*; *Zhang et al., 2009*). TAZ and YAP have emerged as central oncoproteins (*Harvey et al., 2013*) in many different cancers including breast (*Chan et al., 2008*; *Zhao et al., 2012*), colorectal (*Wang et al., 2013*), liver (*Zender et al., 2006*), lung (*Xie et al., 2012*), pancreas (*Zhang et al., 2014*), thyroid (*de Cristofaro et al., 2011*), and sarcomas (*Fullenkamp et al., 2016*; *Merritt et al., 2018*; *Isfort et al., 2019*; *Eisinger-Mathason et al., 2015*; *Crose et al., 2014*). Despite this observation, point mutations within the Hippo signaling pathway are relatively rare in most cancers (*Harvey et al., 2013*). This is especially true of TAZ and YAP, which instead appear to be more commonly mutated by virtue of gene fusions, including *WWTR1*(TAZ)-*CAMTA1* and *YAP1-TFE3*.

TAZ-CAMTA1 transforms cells by driving a predominantly TAZ-related transcriptional program that is mediated by binding to TEAD4 (*Tanas et al., 2016*). Although homologous regions of YAP and TAZ are found in the YAP-TFE3 and TAZ-CAMTA1 fusion proteins, detailed functional studies of YAP-TFE3 have not been performed. Furthermore, the mechanism by which these hybrid transcription factors regulate transcription and subsequent cell transformation has not been defined. Herein we identify the Ada2a-containing histone acetyltransferase (ATAC) complex as a key epigenetic modifier of the TAZ-CAMTA1 and YAP-TFE3 oncogenic transcriptional programs. The ATAC

complex is one of two metazoan complexes that incorporates the GCN5 or closely related PCAF histone acetyltransferase subunit, and primarily acetylates lysine 9 of histone 3 (H3K9) and to a lesser degree lysine 14 (H3K14) (*Nagy et al., 2010*).

There is emerging evidence that the ATAC complex is an oncogenic driver in cancer. YEATS2, one of the subunits of the ATAC complex, has been shown to be functionally important to the pathogenesis of non-small cell lung cancer (*Mi et al., 2017*). Amplification of *YEATS2* has been identified in ovarian cancer, head and neck cancer, esophageal cancer, and uterine cancer (*Mi et al., 2017*), as well as both well-differentiated and dedifferentiated liposarcoma (*Beird et al., 2018*). Herein, we show that the ATAC complex is a key epigenetic mediator of the oncogenic properties of both the TAZ-CAMTA1 and YAP-TFE3 fusion proteins and thereby a potential therapeutic target for essentially all cases of EHE.

## Results

### TAZ-CAMTA1 and YAP-TFE3 are chimeric transcription factors

The *WWTR1* (TAZ)-*CAMTA1* gene fusion fuses the end of exon 2 or exon 3 of *WWTR1* to a breakpoint with exon 9 of *CAMTA1*(*2, 3*). This results in the N terminus of TAZ being fused in frame to the C terminus of CAMTA1. The N terminus of TAZ contains the TEAD binding domain and 14-3-3 protein binding site (*Kanai et al., 2000*). The WW domain may be absent or present, depending on the type of gene fusion. The C terminus of CAMTA1 contributes the transactivating domain, the TIG domain, ankyrin domain, IQ domain, and nuclear localization signal (*Figure 1A*; *Tanas et al., 2016*; *Finkler et al., 2007*; *Long et al., 2014*) . Similarly, *YAP1-TFE3* fuses exon one from *YAP1* in frame to exon 4 of *TFE3* resulting in the N terminus of YAP fused in frame to the C terminus of TFE3. The N terminus of YAP contains the TEAD-binding domain (*Gaffney et al., 2012*; *Figure 1A*). The C terminus of TFE3 contains the transactivating domain, basic helix loop helix domain, and leucine zipper (*Antonescu et al., 2013*; *Kauffman et al., 2014*). TFE3 is a member of the MiTF family of basic helix-loop-helix containing transcription factors, consisting of four structurally similar genes (*TFE3, TFE3B, TFEC,* and *MITF*), that are involved in regulating tissue-specific functions of cell differentiation (*Kauffman et al., 2014*; *Steingrimsson et al., 2002*). TFE3 is ubiquitously expressed and activates transcription by binding to the µE3 motif within the immunoglobulin heavy-chain enhancer (*Beckmann et al., 1990*; *Henthorn et al., 1991*). TFE3 has recently been shown to promote cell survival in starvation conditions by regulating lysosomal biogenesis and homeostasis while also having a role in clearance of cellular debris (*Martina et al., 2014*; *Slade and Pulinilkunnil, 2017*). TFE3 is also involved in gene fusions in renal cell carcinoma and alveolar soft part sarcoma (*Kauffman et al., 2014*; *Argani et al., 2003*; *Klatte et al., 2012*).

### YAP-TFE3 requires components of both YAP and TFE3 to transform cells

In order to dissect the function of the YAP-TFE3 fusion protein, we expressed it in NIH 3T3 cells, which have been utilized to study various components of the Hippo pathway (*Zhao et al., 2007*; *Zhao et al., 2008*; *Zhang et al., 2009*). We also expressed the fusion protein in SW872 cells, a human liposarcoma cell line, chosen because it has an attenuated ability to grow in an anchorage independent manner in either soft agar or poly-HEMA (poly 2-hydroxyethyl methacrylate; data not shown) and because there are no EHE cell lines available. Both TAZ-CAMTA1 and YAP-TFE3 were able to promote colony formation in soft agar in SW872 cells (*Figure 1—figure supplement 2E*). To investigate YAP-TFE3's ability to transform cells, full-length YAP, full-length TFE3, the truncated portions of YAP or TFE3 present in the fusion protein, as well as YAP-TFE3 were expressed in NIH 3T3 cells (*Figure 1—figure supplement 1*) which were then grown in soft agar (*Figure 1B*) and poly-HEMA (*Figure 1C*). YAP-TFE3 promoted anchorage independent growth while full length YAP, full-length TFE3, or the truncated portions of YAP or TFE3 present in the fusion protein did not. Expressing the same constructs in SW872 cells (*Figure 1—figure supplement 1*) and evaluating proliferation on poly-HEMA yielded similar results (*Figure 1D*). The above findings indicate that YAP-TFE3 is a neomorphic protein combining properties of both YAP and TFE3 to transform cells.

YAP and TAZ dynamically shuttle between the cytoplasm and nucleus and this is regulated by Hippo pathway-dependent and -independent mechanisms (*Manning et al., 2020*). Furthermore, the

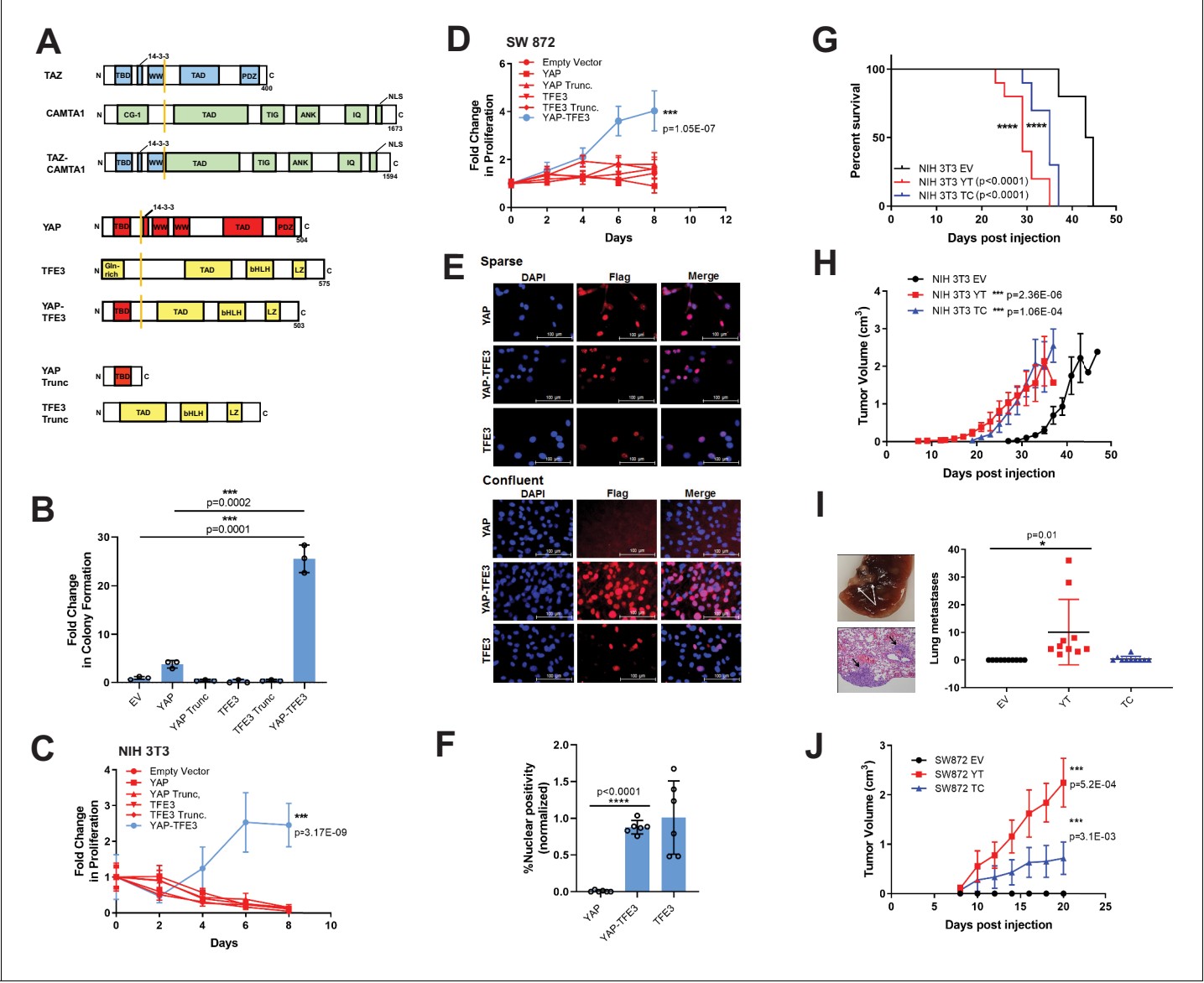

**Figure 1.** YAP-TFE3 transforms cells in vitro and in vivo and combines properties of both YAP and TFE3. (A) Structure of full-length TAZ, CAMTA1, TAZ-CAMTA1 (TC), YAP, TFE3, YAP-TFE3 (YT), truncated portion of YAP (YAP trunc) and the truncated portion of TFE3 (TFE3 trunc). (B) Soft agar assay in NIH 3T3 cells transduced with empty vector (EV), full-length YAP, YAP trunc, TFE3, TFE3 trunc, and YAP-TFE3. (C) Poly-HEMA proliferation assay in NIH 3T3 cells (same constructs as part B). (D) Poly-HEMA proliferation assay in SW872 cells (same constructs as part B). (E) Immunofluorescence images in NIH 3T3 cells expressing YAP, YAP-TFE3, and TFE3 during sparse and confluent conditions, represented graphically (% nuclear positivity during cell confluence is normalized to % positivity in sparse conditions) in (F). (G) Overall survival curve for NSG mice containing xenografted NIH 3T3 cells transduced with empty vector (EV), YAP-TFE3 (YT), and TAZ-CAMTA1 (TC). (H) Tumor growth curve for NSG mice containing xenografted NIH 3T3 cells transduced with empty vector (EV), YAP-TFE3 (YT), and TAZ-CAMTA1 (TC). (I) Gross and microscopic pathology of pulmonary metastases derived from NIH 3T3 cells derived tumors expressing YAP-TFE3. Graphically represented on the right. (J) Tumor growth curve for NSG mice containing xenografted SW872 cells transduced with empty vector (EV), YAP-TFE3 (YT), and TAZ-CAMTA1 (TC). For soft agar assays, statistical significance was evaluated using an unpaired two-tailed t-test. For poly-HEMA proliferation assays, statistical significance was evaluated using fold change increase in proliferation at day 10 with an unpaired two-tailed t-test. For immunofluorescence, % nuclear positivity was calculated for six different fields for each condition; statistical significance was evaluated using an unpaired two-tailed t-test. All in vitro assays were repeated at least twice. Xenograft mouse experiments were repeated twice, using 5–10 mice per group; statistical significance was evaluated using an unpaired two-tailed t-test for evaluation of metastasis and difference in primary tumor size (last day evaluated). Approximately equal numbers of male and female mice were used. Statistical analysis for Kaplan-Meier survival analysis was performed with the log-rank (Mantel-Cox) test. Error bars were used to define one standard deviation. For all panels, ****p<0.0001, ***p<0.001, **p<0.01, *p<0.05.

The online version of this article includes the following figure supplement(s) for figure 1:

*Figure 1 continued on next page*

*Figure 1 continued*

**Figure supplement 1.** YAP-TFE3, TAZ-CAMTA1, and controls are expressed in NIH 3T3 cells, SW 872 cells, and HEK293 cells.

**Figure supplement 2.** YAP-TFE3 is not regulated by the Hippo pathway and required binding to TEAD transcription factors to drive its oncogenic transcriptional program.

**Figure supplement 3.** YAP-TFE3 and TAZ-CAMTA1 driven xenografts in NOD.*Cg-Prkdc^{scid}Il2rg^{tm1Wjl}/SzJ* (NSG) mice form tumors in vivo.

abundance of both YAP and TAZ is limited by ubiquitin-mediated degradation which is controlled by the Hippo pathway (*Liu et al., 2010*; *Huang et al., 2012*). The relative abundance of YAP and TAZ in the nucleus and cytoplasm is modulated by multiple features including cell confluence and mechanotransduction cues (*Manning et al., 2020*; *Zanconato et al., 2016*). Like YAP, we found YAP-TFE3 to be localized within the nucleus when cells were plated under sparse conditions. Under confluent conditions, YAP was enriched in the cytoplasm and its abundance was reduced, while YAP-TFE3 remained predominantly nuclear (*Figure 1E–F*; *Figure 1—figure supplement 2A–B*). Nuclear and cytoplasmic fractionation confirmed that YAP was no longer localized in the nucleus and instead accumulated in the cytoplasm where it was subsequently degraded. In contrast, YAP-TFE3 remained in the nucleus upon cell confluence (*Figure 1—figure supplement 2C*). Similarly, YAP localization shifted from the nucleus to the cytoplasm upon cell detachment, while YAP-TFE3 remained constitutively nuclear (*Figure 1—figure supplement 2D*). Taken together, the above data suggest that YAP-TFE3 is no longer subject to Hippo pathway-mediated suppression, likely because the key residues targeted by LATS1/2 (S127 and S381) (*Zhao et al., 2007*) are not present in YAP-TFE3.

## YAP-TFE3 and TAZ-CAMTA1 drive tumor initiation and proliferation in vivo

To study the function of YAP/TAZ fusion proteins in vivo, NIH 3T3 cells or SW872 cells transduced with empty vector, TAZ-CAMTA1, or YAP-TFE3 were injected into subcutaneous tissue of *Prkdc^{scid}: Il2rg^{tm1Wjl}/SzJ* (NSG) mice. Mice with NIH 3T3-derived tumors expressing YAP-TFE3 or TAZ-CAMTA1 demonstrated a shorter overall survival compared to empty vector (p<0.0001 for both YAP-TFE3 and TAZ-CAMTA1) (*Figure 1G*). Tumors expressing YAP-TFE3 and TAZ-CAMTA1 formed palpable masses ~ 7 days post-injection (*Figure 1H*) indicating time to tumor initiation was similar for NIH 3T3 cells expressing either fusion protein. NIH 3T3 cells containing empty vector formed tumors, as has been described (*Greig et al., 1985*), but tumor initiation was delayed by approximately 10 days. The slopes of the growth curves for the two fusion proteins were similar to empty vector, and this was confirmed by Ki-67 labeling (*Figure 1—figure supplement 3A and B*). Although this is seemingly discrepant with in vitro data showing that YAP-TFE3 drives proliferation (*Figure 1C and D*), the in vitro proliferation being measured was in the context of anchorage independent growth, which may play a more important role in tumor initiation. Mice were sacrificed when tumors reached 2 cm in the greatest dimension (according to animal use protocol). NIH 3T3 cell-derived tumors expressing YAP-TFE3 or TAZ-CAMTA1 exhibited a more rounded (epithelioid) cytomorphology and a greater degree of cellular pleomorphism as compared to control tumors (*Figure 1—figure supplement 3A*). Necropsy and pathological evaluation of the lungs demonstrated the presence of microscopic metastases in the YAP-TFE3 (p=0.0299) but not the TAZ-CAMTA1 (not significant) cohorts (*Figure 1I*). In the mildly tumorigenic SW872 cell line (*Stratford et al., 2012*; *Zhang et al., 2013a*; *Li et al., 2014*), expression of YAP-TFE3 and TAZ-CAMTA1 decreased latency to tumor formation compared to empty vector in vivo (*Figure 1J*), similar to NIH 3T3 cells. Cytological changes in SW872 cell-derived tumors were less pronounced than in NIH 3T3 cells (*Figure 1—figure supplement 3B*). The above data show that both fusion proteins play a role in tumor initiation and can drive tumorigenesis in vivo.

## TEAD transcription factors mediate the oncogenic activity of both YAP-TFE3 and TAZ-CAMTA1

TAZ and YAP are transcriptional co-activators that lack DNA binding domains and form a physical complex with DNA binding transcription factors in order to drive transcription, the best characterized of which are TEAD1-4. TAZ and YAP interact with TEAD1-4 via the TEAD binding domain in the

YAP and TAZ amino termini (*Zhao et al., 2008*; *Zhang et al., 2009*), while the WW domains of YAP and TAZ mediate interactions with other transcription factors including TBX5 (*Murakami et al., 2005*), Runx2 (*Hong et al., 2005*), PPAR-γ (*Hong et al., 2005*), and SMADs (*Varelas et al., 2008*). We previously showed that TEAD4 is required for the oncogenic function of TAZ-CAMTA1 (*Tanas et al., 2016*) . To determine if the WW domain is essential for cell transformation, alanines were substituted for the two essential tryptophans in the WW domain of TAZ-CAMTA1 (TAZ-CAMTA1ΔWW). While the S51A mutant (disrupts TEAD binding) of TAZ-CAMTA1 significantly reduced colony formation in soft agar/growth on poly-HEMA (*Tanas et al., 2016*), the TAZ-CAMTA1ΔWW mutant did not (*Figure 1—figure supplement 2F and G*). This confirms that TAZ-CAMTA1 function is dependent on the TEAD binding domain but not the WW domain.

YAP-TFE3 (*Figure 1A*) contains the TEAD binding domain, but not the WW domain, suggesting that the TEAD binding domain mediates the YAP-TFE3 transcriptional program. To test the hypothesis that TEAD binding is necessary to activate the oncogenic YAP-TFE3 transcriptional program, Tead1 was knocked down in NIH-3T3 cells expressing YAP-TFE3 (*Figure 2A*; *Figure 2—source data 1*). Knock-down of Tead1 reduced anchorage independent growth on poly-HEMA (*Figure 2B*) and in soft agar (*Figure 2C*) with two different short hairpin RNAs (shRNAs). A YAP-TFE3 S94A mutant was created to disrupt its interaction with TEAD (homologous to S94 in full-length YAP and required for TEAD binding *Zhao et al., 2008*, which was confirmed by co-immunoprecipitation (*Figure 2D*)). A prominent band at a lower molecular weight than full-length YAP-TFE3 S94A was identified, likely representing an unstable degradation product. The S94A mutation completely abrogated YAP-TFE3-driven anchorage-independent growth on poly-HEMA (*Figure 2E*) and in soft agar (*Figure 2F*). By quantitative RT-PCR, the YAP-TFE3 S94A mutant showed a sevenfold decrease in expression of *Ccn2*(Ctgf), a YAP target gene, compared to the YAP-TFE3 control (*Figure 2G*). Chromatin immuno-precipitation-quantitative PCR for the *CCN2* promoter (which contains the TEAD-binding sequence) showed a sixfold decrease in *CCN2* promoter binding by YAP-TFE3 S94A as compared to YAP-TFE3 control (*Figure 2H*). Using the 8XGTIIC luciferase reporter containing eight TEAD consensus binding sequences (*Dupont et al., 2011*), YAP-TFE3 S94A showed decreased luciferase activity compared to the YAP-TFE3 control (*Figure 2I* and *Figure 1—figure supplement 1E*). Collectively, these data suggest that YAP-TFE3, like TAZ-CAMTA1, requires TEAD transcription factors to drive its oncogenic transcriptional program (*Figure 2J*).

## The fusion proteins induce overlapping but different transcriptomes than TAZ and YAP

From this working model, we hypothesized that since TEAD binding is essential for YAP-TFE3 and TAZ-CAMTA1-driven transformation, the YAP-TFE3 and TAZ-CAMTA1 transcriptomes would essentially recapitulate the YAP and TAZ transcriptional programs, respectively. To test this hypothesis, RNA-Seq was performed on NIH 3T3 cells and SW872 cells stably expressing TAZ4SA (hyperactivated form of TAZ due to alanines being substituted for four serines that can be phosphorylated by the LATS1/2 kinases [*Lei et al., 2008*]), TAZ-CAMTA1, CAMTA1, YAP5SA (hyperactivated form of YAP due to alanines being substituted for five serines that can be phosphorylated by the LATS1/2 kinases [*Zhao et al., 2007*]), YAP-TFE3, and TFE3, as well as cells transduced with empty vector control (*Figure 1—figure supplement 1*; *Figure 3—source data 1* and *2*). Differentially expressed genes (FDR < 0.05) showed considerable overlap between the TAZ4SA and TAZ-CAMTA1 transcriptomes; 70% of genes differentially expressed in the TAZ-CAMTA1 transcriptome were also present in the TAZ4SA transcriptome with a $-\log_{10}$[hypergeometric density] (HGD) of 105.6 ($p=1.92\times10^{-106}$). By comparison, a much smaller percentage of overlap (9%) was identified between the TAZ-CAMTA1 and CAMTA1 gene sets with an HGD of 34.1 ($p=4.71\times10^{-35}$). A significant percentage (29%) of differentially expressed genes in the TAZ-CAMTA1 transcriptional program were not induced by either TAZ4SA or CAMTA1 (*Figure 3A*; *Figure 3—source data 1*) in NIH 3T3 cells. Similarly, 31% of the TAZ-CAMTA1 gene set in SW872 cells were unique and not found in the TAZ or CAMTA1 transcriptional programs (*Figure 3B*; *Figure 3—source data 2*). As in NIH 3T3 cells, the amount of overlap between TAZ-CAMTA1 and TAZ4SA in SW872 cells was greater (67% of TAZ-CAMTA1 genes; HGD of 72.6) than that between TAZ-CAMTA1 and CAMTA1 (9% of TAZ-CAMTA1 genes; HGD of 27.6).

The YAP-TFE3 transcriptional program demonstrated similar findings. In NIH 3T3 cells, 78% of genes differentially expressed in the YAP-TFE3 transcriptome were present in the YAP5SA

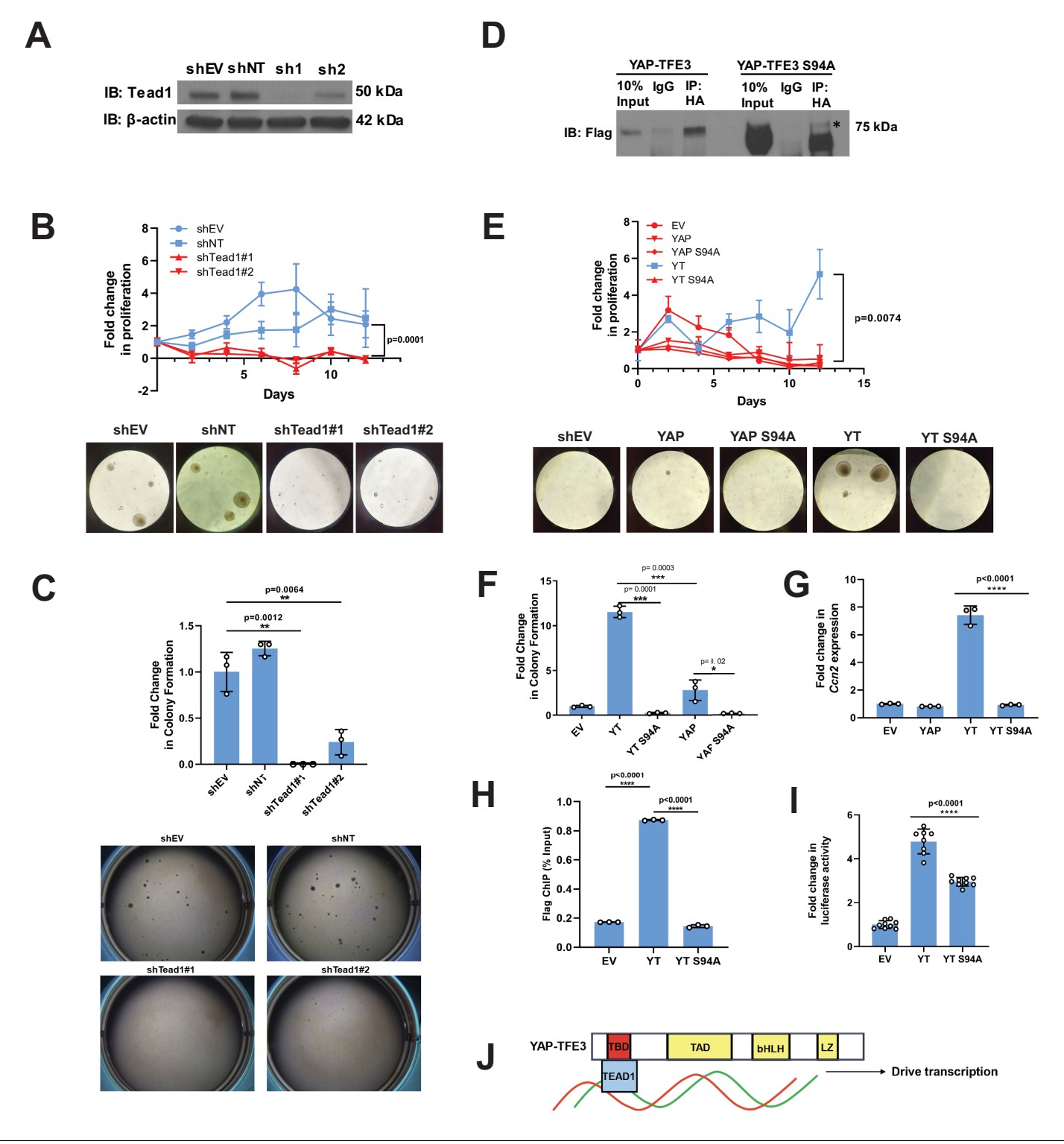

**Figure 2.** Tead1 mediates the oncogenic transcriptional program of YAP-TFE3. (**A**) Knockdown of Tead1 in NIH 3T3 YAP-TFE3 cells. (**B**) Poly-HEMA analysis in NIH 3T3 YAP-TFE3 cells with Tead1 knock-down. Spheroid formation on poly-HEMA is shown below. (**C**) Soft agar assay in NIH 3T3 YAP-TFE3 cells with two Tead1 shRNA (shTead#1 and shTead#2) with colonies shown below. (**D**) Co-immunoprecipitation experiment in NIH 3T3 cells stably expressing Flag-YAP-TFE3 or Flag-YAP-TFE3 S94A. 3HA-TEAD1 is transiently transfected. * indicates MW of Flag-YAP-TFE3. (**E**) Poly-HEMA assay in NIH 3T3 cells expressing same constructs present in E. Spheroid formation on poly-HEMA is shown below. (**F**) Soft agar assay in NIH 3T3 cells expressing YAP-TFE3 (YT), YT S94A, YAP, or YAPS94A. (**G**) Quantitative RT-PCR showing reduced *Ccn2* expression in NIH 3T3 YAP-TFE3 (YT) S94A cells. *Figure 2 continued on next page*

*Figure 2 continued*

(H) Chromatin immunoprecipitation-quantitative PCR for the *CCN2* promoter in SW872 cells transduced with empty vector (EV), YT, and YT S94A. (I) Luciferase reporter assay (8XGTIIC luciferase reporter construct) in HEK 293 cells expressing YT or YT S94A. (J) Model diagram of YAP-TFE3 driving transcription by binding to TEAD1. For soft agar assays, statistical significance was evaluated using an unpaired two-tailed *t*-test. For poly-HEMA proliferation assays, statistical significance was evaluated using fold change increase in proliferation at day 10 with an unpaired two-tailed *t*-test. For quantitative RT-PCR and the luciferase reporter assay, standard deviation was calculated from fold change values for each triplicate. Each experiment was repeated at least twice. Error bars were used to define one standard deviation. For all panels, ****p<0.0001, ***p<0.001, **p<0.01, *p<0.05. The online version of this article includes the following source data for figure 2:

**Source data 1.** Tead1 shRNA sequences.

transcriptional program. Comparatively, only 10% of differentially expressed genes in the YAP-TFE3 gene set were also present in the TFE3 transcriptome. In NIH 3T3 cells 20% of the differentially expressed genes were unique to the YAP-TFE3 transcriptome and not found in the YAP5SA or TFE3 transcriptional programs (*Figure 3C*; *Figure 3—source data 1*). Similarly, in SW872 cells 47% of the YAP-TFE3 transcriptional program was found only in the YAP-TFE3 transcriptome and not in YAP5SA or TFE3 controls (*Figure 3D*; *Figure 3—source data 2*). Mirroring findings in NIH 3T3 cells, the amount of overlap between YAP-TFE3 and YAP5SA (53% of YAP-TFE3 genes; HGD of 220.1) was greater than that between YAP-TFE3 and TFE3 (0.2% of YAP-TFE3 genes; HGD of 1.5).

A principal component analysis (PCA) revealed that the TAZ-CAMTA1 and YAP-TFE3 transcriptomes in NIH 3T3 cells grouped more closely together than the transcriptomes of their cognate full length proteins (*Figure 3E*). In SW872 cells, the TAZ-CAMTA1 and YAP-TFE3 transcriptomes diverged from one another, but were still distinct from the TAZ4SA and YAP5SA transcriptomes (*Figure 3—figure supplement 1A*), providing further support that the fusion proteins drive a transcriptional program that incorporates some elements of the TAZ/YAP transcriptomes but is also significantly altered from the baseline TAZ and YAP transcriptional program. This was corroborated by qRT-PCR experiments, which showed that TAZ-CAMTA1 and YAP-TFE3 promote upregulation of the well-defined TAZ/YAP transcriptional targets *CCN2*(CTGF)/*Ccn2* and *CCN1*(CYR61)/*Ccn1*, but also genes unique to the TAZ-CAMTA1 and YAP-TFE3 transcriptomes such as *Fras1* (an extracellular matrix protein), *Erbb3* (Erb-B2 Receptor Tyrosine Kinase 3) (*Figure 3F*), and *FBLN5* (an extracellular matrix protein) (*Figure 3G*). iPathwayGuide analysis performed showed that the top cancer-related signatures activated by TAZ-CAMTA1 in SW872 cells (*Figure 3H*) are the PI3K-Akt signaling pathway, Hippo signaling pathway (*Figure 3—figure supplement 1B*), focal adhesion, proteoglycans, and extracellular matrix (ECM)-receptor interaction (mirroring qRT-PCR results in *Figure 3F and G*). The top cancer-related signatures activated by YAP-TFE3 in SW872 cells (*Figure 3I*) also included the PI3K-Akt signaling pathway, as well as the MAPK and JAK-STAT signaling pathways. A microRNA and apoptosis signature were also identified.

## The TAZ-CAMTA1 and YAP-TFE3 chromatin binding profiles are enriched for both TEAD and non-TEAD transcription factor motifs

To determine if the modified transcriptional programs of the fusion proteins were driven by altered DNA binding, we performed chromatin immunoprecipitation (ChIP) sequencing studies on SW872 cells using Flag-tagged TAZ4SA, TAZ-CAMTA1, CAMTA1, YAP5SA, YAP-TFE3, and TFE3 constructs. Annotation of ChIP peaks with respect to their nearest gene transcriptional start site (TSS) demonstrated that CAMTA1 and TFE3 ChIP peaks were localized in closer proximity to the TSS (0–1 kb). In contrast, TAZ4SA, TAZ-CAMTA1, YAP5SA, and YAP-TFE3 predominantly occupied distal intergenic sequences (3 kb or greater) consistent with active enhancer sequences (*Figure 4A–E*; *Figure 4—figure supplement 1A–D*) as confirmed by the presence of overlapping H3K27ac ChIP peaks (*Zhang et al., 2013b*; *Creyghton et al., 2010*; *Calo and Wysocka, 2013*) (69% overlap in TAZ-CAMTA1 distal intergenic sequences; 68% overlap in YAP-TFE3 distal intergenic sequences) (*Figure 4D and E*). Although the fusion proteins most frequently bound to distal intergenic sequences/active enhancers, they also bound to sequences closer to the TSS (e.g. promoters regions ≤ 1 kb from the TSS) (*Figure 4D and E*), which are also enriched for H3K27ac marks (*Creyghton et al., 2010*). Comparison of TAZ-CAMTA1 and H3K27ac genomic occupancy around the TSS of *MAFK* and *HOXA1* (*Figure 4—figure supplement 1E,F*), two differentially expressed genes in TAZ-CAMTA1-expressing cells (*Figure 3—source data 2*), showed that H3K27ac peaks closely

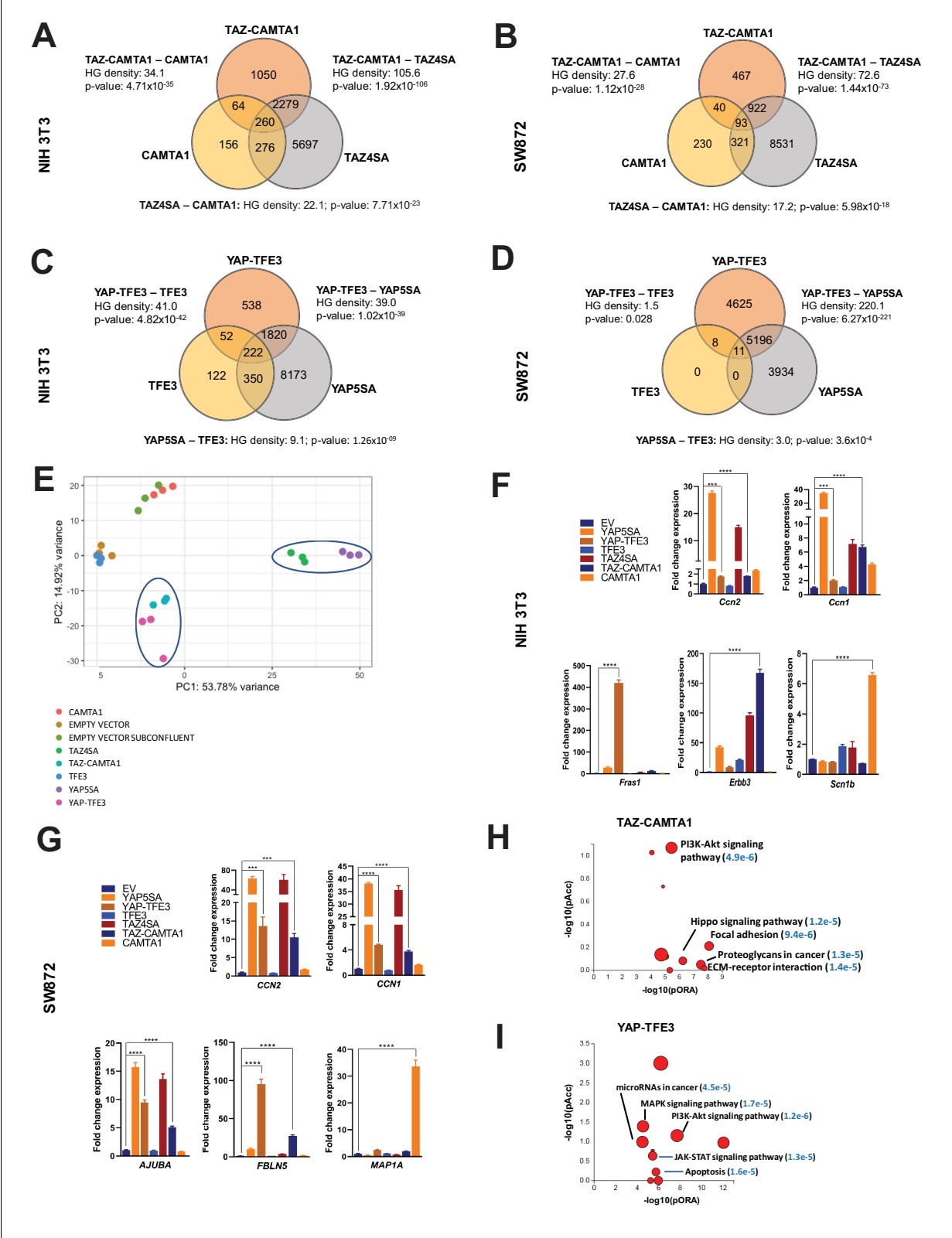

**Figure 3.** TAZ-CAMTA1 and YAP-TFE3 induce different but overlapping transcriptomes compared to TAZ and YAP. (**A**) Number of differentially expressed genes (FDR = 5%) in NIH 3T3 cells expressing TAZ-CAMTA1 and full-length controls. HG density represents the –log$_{10}$(hypergeometric density). p Value for hypergeometric analysis included. (**B**) Number of differentially expressed genes (FDR = 5%) in SW872 cells expressing TAZ-CAMTA1 and full-length controls. (**C**) Number of differentially expressed genes (FDR = 5%) in NIH 3T3 cells expressing YAP-TFE3 and full-length

*Figure 3 continued on next page*

*Figure 3 continued*

controls. (D) Number of differentially expressed genes (FDR = 5%) in SW872 cells expressing YAP-TFE3 and full-length controls. (E) Principal component analysis of RNA expression after variance-stabilizing transformation in NIH 3T3 cells. (F) Validation of RNA-Seq in NIH 3T3 cells by qRT-PCR for key genes. (G) Validation of RNA-Seq in SW872 cells by qRT-PCR for key genes. (H) Scatter plot of two types of pathway enrichment evidence: probability of over-representation (pORA) and probability of accumulation (pAcc) as calculated by iPathwayGuide for SW872 TAZ-CAMTA1 cells and (I) YAP-TFE3 cells. RNA-Seq experiments in NIH 3T3 cells and SW872 cells were performed using biological triplicates for each of the conditions (expression constructs). For gene expression data, the population was set to the total number of recovered genes with the mean of 5 counts across all samples. Hypergeometric testing was performed using the phyper() function in the stats R package (v3.6.3) set to assess enrichment and the lower tail set to false. Hypergeometric density was calculated using the related dhyper function and converted using the negative log10 of the output. For quantitative RT-PCR, standard deviation was calculated from fold change values for each triplicate. Error bars were used to define one standard deviation. For all panels, ****p<0.0001, ***p<0.001, **p<0.01, *p<0.05.

The online version of this article includes the following source data and figure supplement(s) for figure 3:

**Source data 1.** Differentially expressed genes in NIH 3T3 cells expressing fusion proteins and full length controls.
**Source data 2.** Differentially expressed genes in SW872 cells expressing fusion proteins and full length controls.
**Figure supplement 1.** Differentially expressed genes in SW872 cells expressing fusion proteins and full length controls.

overlapped and surrounded TAZ-CAMTA1 bound chromatin, validating the above approach. Thus TAZ-CAMTA1 and YAP-TFE3 maintain the genomic occupancy characteristics of increased preference for distal intergenic/enhancer sequences as demonstrated by TAZ and YAP (*Figure 4—figure supplement 1A,C*; *Zanconato et al., 2015*; *Galli et al., 2015*; *Stein et al., 2015*), but also occupy sequences proximal to the TSS.

ChIP peaks were annotated to their nearest gene features (as described in Materials and methods). Mirroring the RNA-Seq data, 56% of TAZ-CAMTA1 genes containing ChIP peaks were present in the TAZ-CAMTA1 data set, but not the TAZ4SA or CAMTA1 data sets (*Figure 4F*). Similarly, 46% of YAP-TFE3 genes containing ChIP peaks were unique to the YAP-TFE3 data set and not present in the YAP5SA or TFE3 controls (*Figure 4G*). Overall, the chromatin binding profile of TAZ-CAMTA1 more closely resembled TAZ4SA (HGD of 52.9) than CAMTA1 (HGD of 43.8). Likewise, YAP-TFE3 bound chromatin was more like that of YAP5SA (HGD of 166.8) than TFE3 (HGD of 11.8). Unbiased motif enrichment analysis (*Figure 4—source data 1*) showed that approximately the same proportion of TAZ4SA and TAZ-CAMTA1 peaks contained the TEAD binding sequence (58% vs. 61%, respectively) (*Figure 4F*). Interestingly, there was a decrease in the frequency of FOS/JUN binding sites in TAZ-CAMTA1 (31%) as compared to TAZ4SA (50%), which appeared to be offset by an increase in Early growth response gene-2 (EGR2) binding motifs (23% vs. 13% for TAZ4SA). EGR2 is a transcription factor important to neural development (*Warner et al., 1998*) as is CAMTA1 (*Long et al., 2014*).

Motif enrichment analysis showed a decrease in the fraction of YAP-TFE3-bound peaks containing the TEAD-binding sequence (38%) as compared to YAP5SA-bound peaks (51%) (*Figure 4G*). Microphthalmia-associated transcription factor (MiTF) is a member of the TFE/MiTF transcription factor family known to complex with TFE3 (*Steingrimsson et al., 2002*). Interestingly, the MiTF binding motif was enriched in YAP-TFE3 (58%) compared to YAP5SA (16%) (*Figure 4G*), suggesting that, like TAZ-CAMTA1, TFE3 confers DNA binding properties to the YAP-TFE3 protein in addition to events mediated by YAP's interaction with TEAD transcription factors. Consistent with this, we found that YAP-TFE3 indeed co-immunoprecipitates with TFE3, and that disruption of the bHLH domain in the TFE3 portion of YAP-TFE3 that is responsible for dimerization altered anchorage-independent growth (*Figure 4—figure supplement 2A*). Overall, the motif enrichment analysis explains, at least in part, our observation that the chromatin binding profiles of the fusion proteins overlap significantly, but also differ from, full-length TAZ and YAP. Genes bound by the fusion proteins or controls were additionally validated by ChIP-qPCR (*Figure 4H*).

To determine if genes bound by the fusion proteins were differentially expressed, a plot was generated that arrayed differentially expressed genes in terms of increasing statistical significance along the Y axis, and genes in terms of increasing ChIP peak 'score' (i.e. statistical significance) along the X axis for TAZ-CAMTA1, YAP-TFE3, and controls (*Figure 4—figure supplement 2B–F*; *Figure 4—source data 2*). This analysis revealed well known YAP/TAZ target genes such as *CCN1* (CYR61), *CCN2* (CTGF), *AMOTL2*, *AJUBA*, *LATS2*, and *TEAD1*, and genes that have not been identified as YAP/TAZ targets, such as *MAFK*. Thirteen percent of differentially expressed genes for TAZ-

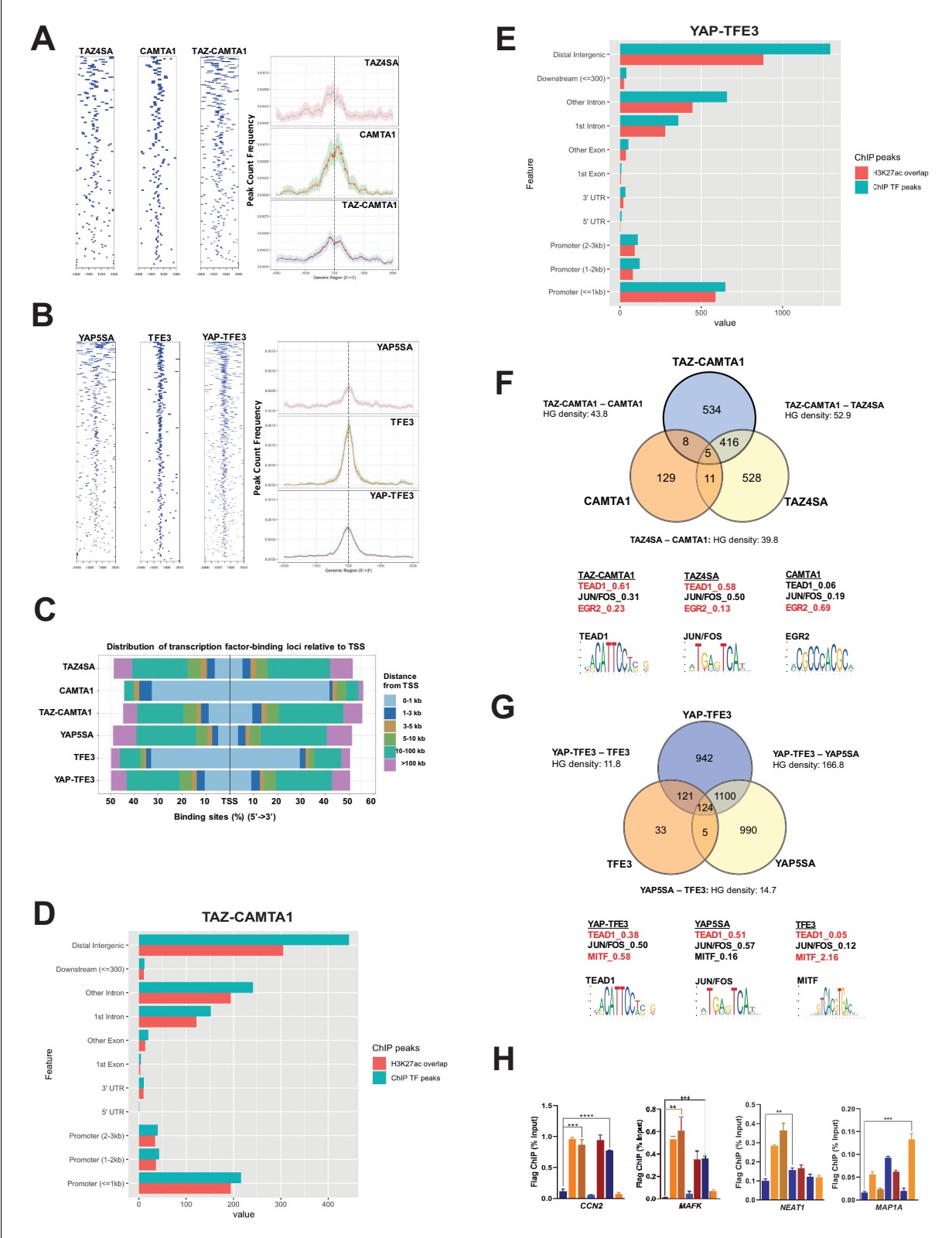

**Figure 4.** TAZ-CAMTA1 and YAP-TFE3 occupy TEAD and non-TEAD transcription factor motifs. (**A**) Heat map of ChIP binding to transcriptional start site (TSS) regions and average TSS profile histogram for TAZ-CAMTA1 and controls. (**B**) Heat map and histogram for YAP-TFE3 and controls. (**C**) Distribution of transcription factor binding loci relative to TSS for the fusion proteins and controls. (**D**) Distribution of overlapping TAZ-CAMTA1 and H3K27ac ChIP peaks among annotated functional DNA-binding sites. (**E**) Distribution of overlapping YAP-TFE3 and H3K27ac ChIP peaks among

*Figure 4 continued on next page*

*Figure 4 continued*

annotated functional DNA binding sites. (F) Intersection of gene annotations for TAZ-CAMTA1 ChIP peaks and controls. Proportion of peaks containing selected transcription factor (TF) motifs (consensus sequences included) shown for each construct below. HG density represents the $-\log_{10}$(hypergeometric density). (G) Intersection of gene annotations for YAP-TFE3 ChIP peaks and controls. Proportion of peaks containing selected TF motifs shown for each construct below. (H) Validation of ChIP-Seq in SW872 cells in selected genes by ChIP-qPCR. ChIP-Seq experiments in SW872 cells were performed using biological triplicates for each of the conditions (expression constructs). For ChIP-Seq analysis, the population was set as the total number of genes annotated across all conditions. Hypergeometric testing was performed using the phyper() function in the stats R package (v3.6.3) set to assess enrichment and the lower tail set to false. Hypergeometric density was calculated using the related dhyper function and converted using the negative log10 of the output. For ChIP-qPCR, standard deviation was calculated from fold change values for each triplicate. Error bars were used to define one standard deviation. For all panels, ****p<0.0001, ***p<0.001, **p<0.01, *p<0.05.

The online version of this article includes the following source data and figure supplement(s) for figure 4:

**Source data 1.** ChIP AME Motif Enrichment Analysis.
**Source data 2.** Differentially expressed genes also bound by fusion proteins and controls as determined by ChIP-Seq.
**Figure supplement 1.** Distribution of overlapping transcription factor and H3K27ac ChIP peaks among annotated functional DNA-binding sites.
**Figure supplement 2.** Characterization of YAP-TFE3 and TAZ-CAMTA1 DNA binding in SW872 cells.
**Figure supplement 3.** TAZ-CAMTA1 and YAP-TFE3-bound genes are differentially expressed.

CAMTA1 were directly bound by the transcription factor (*Figure 4—figure supplement 3A*). Similarly, 15% of differentially expressed genes for YAP-TFE3 were directly bound by the fusion protein (*Figure 4—figure supplement 3B*). Kyoto encyclopedia of genes and genomes (KEGG) analysis of TAZ-CAMTA1 bound genes that are also differentially expressed in SW872 TAZ-CAMTA1 cells showed enrichment for RHO GTPase-dependent pathways, EPHA-dependent pathways, RUNX2-dependent pathways, as well as other pathways known to be regulated by TAZ/YAP (*Hong et al., 2005*; *Dupont et al., 2011*; *Edwards et al., 2017*; *Figure 4—figure supplement 3C*).

## TAZ-CAMTA1 and YAP-TFE3 augment the euchromatin landscape

Previous studies indicated that TAZ and YAP form physical complexes with multiple different with chromatin modifying proteins (*Moya and Halder, 2014*). To test the hypothesis that the fusion proteins modify chromatin structure, Assay for Transposase-Accessible Chromatin (ATAC)-sequencing studies were performed on SW872 cells expressing the above constructs. Heat map and histogram analysis of ATAC peaks showed that TAZ-CAMTA1 and CAMTA1 preferentially opened chromatin in areas of the genome more distal to the TSS than TAZ4SA (*Figure 5A*, *Figure 5—figure supplement 1A,B*). Similarly, although less pronounced, YAP-TFE3 preferentially opened chromatin architecture more distal to the TSS than YAP5SA (*Figure 5B* and *Figure 5—figure supplement 1A,B*). These results are in keeping with the genomic occupancy profiles of the fusion proteins defined by ChIP-Seq (*Figure 4C–E*) which were predominantly bound to enhancer sequences.

Analysis of differentially accessible chromatin annotated to the nearest gene feature (as described in Materials and methods; note that multiple ATAC peaks could be assigned to the same gene) (*Figure 5C,D*, and *Figure 5—figure supplement 1C,D*) showed that 15,779 genes with annotated ATAC peaks were shared between TAZ-CAMTA1 and all of the controls while 16,809 genes were shared between YAP-TFE3 and all the controls, representing the baseline levels of transposase accessible chromatin. We initially hypothesized that the accessible chromatin landscape for the fusion proteins would overwhelmingly resemble that of full length TAZ and YAP. However, hypergeometric analysis showed that the accessible chromatin profile of TAZ-CAMTA1 most closely resembled that of CAMTA1 (*Figure 5C*), while the $-\log_{10}$(hypergeometric density) HGD between YAP-TFE3 and YAP5SA (106.0) was roughly equivalent to that between YAP-TFE3 and TFE3 (98.0) (*Figure 5D*). This analysis (*Figure 5C,D*) also showed that both TAZ-CAMTA1 and YAP-TFE3 promoted chromatin accessibility for unique genes not found in the control data sets, making them available for either transcriptional activation or repression. More transposase accessible, non-baseline genes were present with expression of TAZ-CAMTA1 (3147 genes) and YAP-TFE3 (2499 genes) than TAZ4SA (656 genes) and YAP5SA (1629 genes), respectively. 'Diffbind' differential binding affinity analysis of consensus-derived peak sets was carried out with respect to empty vector controls. Dimensionality reduction with PCA showed that TAZ-CAMTA1 and YAP-TFE3 samples clustered together with CAMTA1 and TFE3, suggesting a greater degree of similarity in ATAC peak affinity patterns, while TAZ4SA or YAP5SA were distinct and separate (*Figure 5E*). These results are

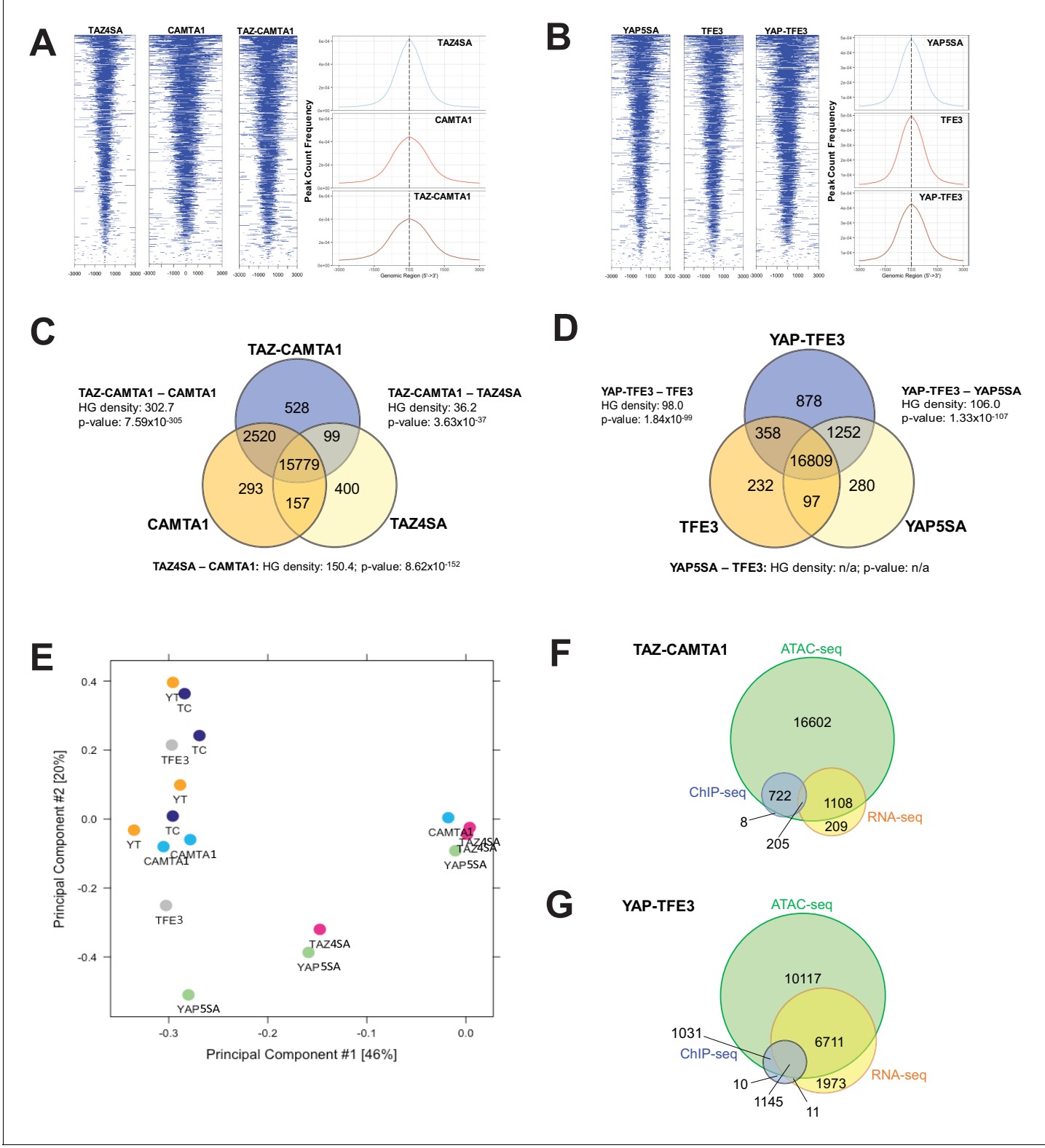

**Figure 5.** TAZ-CAMTA1 and YAP-TFE3 promote the formation of chromatin accessible to transcriptional regulation. (A) Heat map of ATAC peak distribution around the transcriptional start site (TSS) region and average ATAC TSS profile for TAZ-CAMTA1 and controls and (B) for YAP-TFE3 and controls. (C) Intersection of gene annotations for transposase accessible chromatin peaks in TAZ-CAMTA1 and controls and (D) YAP-TFE3 and controls. HG density represents the −log₁₀(hypergeometric density). p value for hypergeometric analysis included. (E) DiffBind principal component analysis for fusion proteins and controls. (F) Overlay of differentially expressed genes, bound genes, with transposase accessible chromatin for TAZ-CAMTA1. (G)

*Figure 5 continued on next page*

*Figure 5 continued*

Overlay of differentially expressed genes, bound genes, with transposase accessible chromatin for YAP-TFE3. ATAC-Seq experiments in SW872 cells were performed using biological triplicates for each of the conditions (expression constructs). For ATAC-Seq analysis, the population was set as the total number of genes annotated across all conditions. Hypergeometric testing was performed using the phyper() function in the stats R package (v3.6.3) set to assess enrichment and the lower tail set to false. Hypergeometric density was calculated using the related dhyper function and converted using the negative log10 of the output.

The online version of this article includes the following figure supplement(s) for figure 5:

**Figure supplement 1.** TAZ-CAMTA1 and YAP-TFE3 alter the chromatin landscape.

largely consistent with the hypergeometric analysis mentioned above. The homology of the resultant chromatin landscape for the two fusion proteins, despite the structural dissimilarity of the C termini of CAMTA1 and TFE3, independently supports a convergent function for the two fusion proteins. Integrating annotation results from all three NGS modalities (RNA-, ChIP-, and ATAC-seq) showed that the majority of genes bound by the fusion proteins, as well as those which were differentially expressed, were within euchromatin regions of the genome (*Figure 5F,G*). To further evaluate whether the fusion protein DNA-binding sites were nested within transposase accessible chromatin, we evaluated the combined ChIP-Seq and ATAC-Seq data for *MAFK* and *HOXA1*, two genes that were bound by one or both of the fusion proteins and simultaneously differentially expressed (*Figure 4—figure supplement 2C,F*). Evaluating the combined ChIP/ATAC signal tracks for *MAFK* and *HOXA1* in TAZ-CAMTA1 and YAP-TFE3 expressing cells (*Figure 5—figure supplement 1E,F*) suggested that the fusion proteins combine the DNA-binding activity of the TEAD domains of TAZ/YAP with the chromatin remodeling properties of CAMTA1 and TFE3 for these genes.

## The TAZ-CAMTA1 and YAP-TFE3 interactomes are enriched for transcriptional regulatory proteins

To address the hypothesis that CAMTA1 and TFE3 confer altered chromatin remodeling properties to TAZ-CAMTA1 and YAP-TFE3, we utilized proximity-dependent biotinylation (BioID) and tandem mass spectrometry. TAZ-CAMTA1, YAP-TFE3, and full-length controls were fused on their N terminus to BirA*, an abortive biotin ligase, in a tetracycline-inducible vector, and stably transfected in Flp-In T-Rex HEK 293 cells. Proteins in proximity to the BirA*-tagged bait proteins were biotinylated, purified, and analyzed by tandem mass spectrometry and SAINT statistical analysis (*Figure 6A*). Prey proteins detected with at least 10 spectral counts in at least one biological replicate were represented in dot plot format as a composite of the relative abundance and spectral count, where the size of the dots represents the relative abundance and darkness represents the spectral count (the false discovery rate is represented as the color coding at the periphery of the dot). Sixty-eight prey proteins were identified that established high-confidence proximity interactions with the fusion proteins or the full-length controls (*Figure 6—figure supplement 1A–E*). The YAP-TFE3 proximity interactome was composed predominantly of histone modifying enzymes (35%), transcriptional regulators (26%), proteins involved in chromatin remodeling (18%), and transcription factor/co-activators (15%) (*Figure 6B*). Similarly, the TAZ-CAMTA1 proximity interactome was composed predominantly of histone modifying enzymes (27%), transcriptional regulators (18%), transcription factor/co-activators (15%), and chromatin remodeling (13%). A smaller percentage of the TAZ-CAMTA1 interactome was involved in mediating cell-cell junctions (5%) and SUMO processing (4%). To identify proteins to prioritize for further study, we subtracted proteins that showed a decreased proximity interaction with the fusion proteins relative to full-length TAZ and YAP; many of which (e.g. MPDZ, PATJ) are part of the Crumbs complex, and the decreased interaction can be explained in part by loss/mitigation of the WW domain. This left 55 and 34 prey proteins in the TAZ-CAMTA1 and YAP-TFE3 interactomes, respectively. Known components of the Hippo pathway were subtracted from further analysis, leaving 49 proteins in the TAZ-CAMTA1 data set, and 31 proteins in the YAP-TFE3 data set. We then retained only prey proteins that demonstrated an increased interaction with the fusion proteins relative to full length TAZ and YAP, leaving 47 proteins in the TAZ-CAMTA1 data set and 28 remaining proteins in the YAP-TFE3 data set. The two data sets were intersected to demonstrate 27 shared proteins (*Figure 6C–E*). As shown in the table listing the function of the shared proteins (*Figure 6D*) and a chart giving an overview of their composition (*Figure 6E*), the shared prey

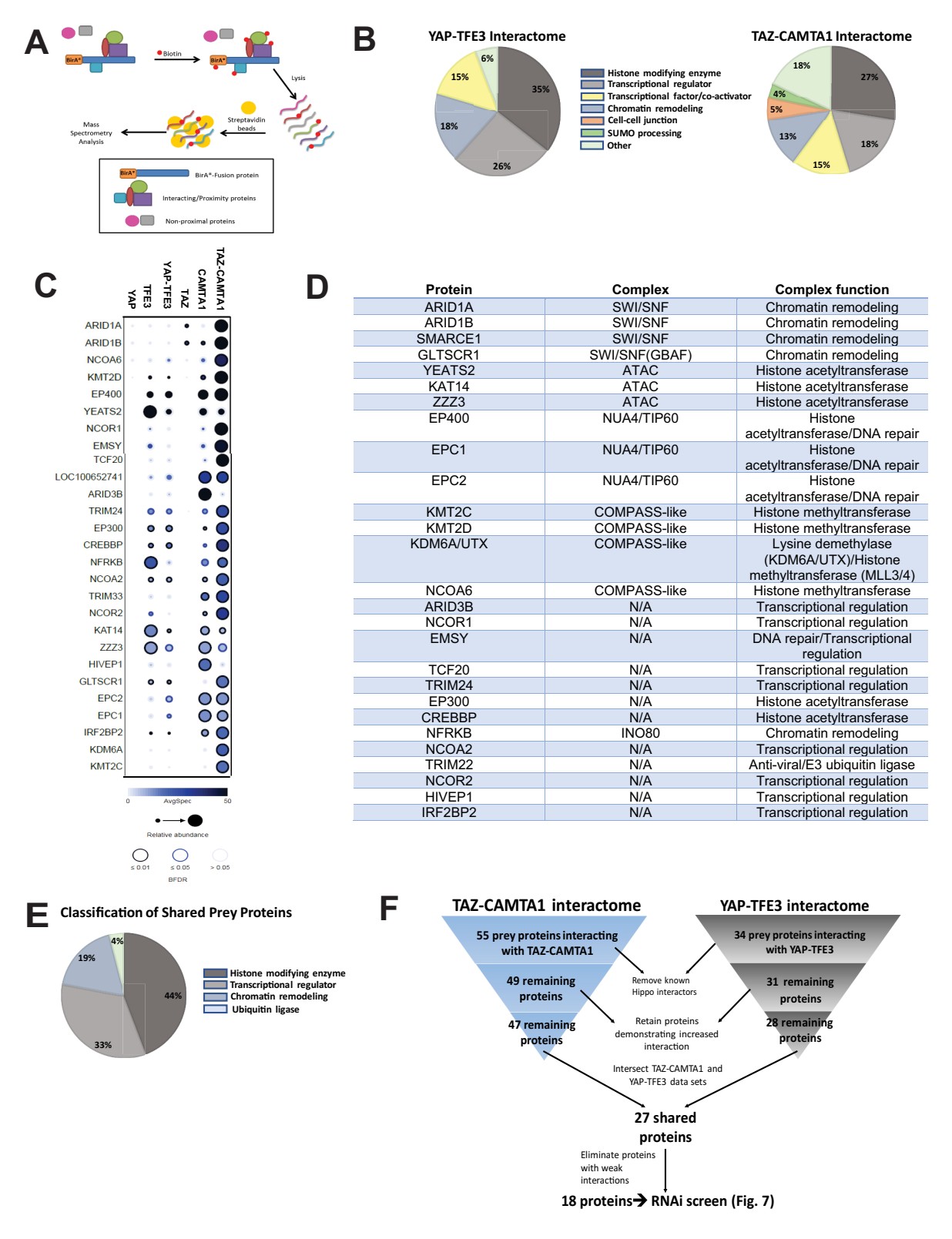

**Figure 6.** The TAZ-CAMTA1 and YAP-TFE3 interactomes are enriched for transcriptional regulators and chromatin modifiers. (**A**) Overview of BioID method. (**B**) Summary of YAP-TFE3 and TAZ-CAMTA1 interactomes. (**C**) Dot plot representation of chromatin modifiers shared by YAP-TFE3 and TAZ-CAMTA1 interactomes. (**D**) Table of chromatin modifiers shared by YAP-TFE3 and TAZ-CAMTA1 interactomes. (**E**) Classification of prey proteins shared between TAZ-CAMTA1 and YAP-TFE3 interactomes. (**F**) Algorithm to prioritize the TAZ-CAMTA1 and YAP-TFE3 interactomes for the subsequent RNAi

*Figure 6 continued on next page*

*Figure 6 continued*

screen. For BioID mass spectrometry, two biological replicates were made for each cell line. Affinity purification and proximity biotinylation coupled to mass spectrometry were performed as described in *Lambert et al., 2015*. SAINT (significance analysis of interactome) analysis (*Choi et al., 2011*) was performed on the mass spectrometry data, using 10 controls compressed to 5. Only proteins with iProphet protein probability ≥ 0.95 were used. Results are expressed in dotplot format. Each prey protein is represented as a dot, with color signifying average spectral count, the darkness indicating average spectral count between the two biological replicates (the darker the dot, the higher the average spectral count), and the size represents the relative abundance. Darkness of the ring indicates the Bayesian False Discovery Rate (BFDR); black (FDR ≤ 0.01), blue (FDR ≤ 0.05), light blue (FDR > 0.05). The data was filtered so that each prey had a minimum of 10 spectral counts in at least one of the biological replicates.

The online version of this article includes the following figure supplement(s) for figure 6:

**Figure supplement 1.** BioID Mass Spectrometry shows that YAP-TFE3 and TAZ-CAMTA1 have an altered interactome compared to YAP and TAZ.

proteins were enriched for epigenetic/transcriptional regulators. Proteins with weak interactions with both fusion proteins (generally these were in the lowest category of spectral count/relative abundance and FDR > 5%) were eliminated, leaving 18 proteins (*Figure 6F*). The observation that so many proximity interactions were shared between TAZ-CAMTA1 and YAP-TFE3, despite having very different C-termini, indicated these shared prey proteins may be important to the oncogenic mechanism of TAZ-CAMTA1 and YAP-TFE3.

## The ATAC complex mediates the transforming potential of TAZ-CAMTA1

To further investigate these 18 prey proteins, an RNAi screen was performed on SW872 cells expressing TAZ-CAMTA1 using five short hairpin constructs (shRNA) for each gene/protein (*Figure 7—source data 1*). These lines were subsequently evaluated by poly-HEMA assay (*Figure 7A*) utilizing the observation that TAZ-CAMTA1 promoted anchorage independent growth in SW872 cells on poly-HEMA (*Figure 7B*) and in soft agar (*Figure 1—figure supplement 2E*). The number of shRNAs that demonstrated a deficit in anchorage independent growth as compared to shEV and shNT were recorded to rank order the genes/proteins in the interactome for further study (*Figure 7C*). Genes that demonstrated five shRNAs promoting a deficit in anchorage independent growth included *YEATS2* (*Figure 7C,D*), *ZZZ3* (*Figure 7C,E*), and *EP400* (*Figure 7C,F*). Other genes/prey proteins with more than two shRNAs showing a phenotype included *NCOA2* (*Figure 7G*) and *TCF20* (*Figure 7H*). Remaining prey proteins/genes with at least one shRNA showing a phenotype are also shown (*Figure 7C*; *Figure 7—figure supplement 1A–F*). *YEATS2* knock-down and a reduction in anchorage independent growth was validated in SW872 cells expressing YAP-TFE3 using the two shRNAs with the best knock-down (*Figure 7—figure supplement 1G–I*).

Significantly, *YEATS2* and *ZZZ3*, the two top ranking hits in the RNAi screen, are both members of the Ada2a-containing histone acetyltransferase complex (ATAC). YEATS2 is the scaffolding protein for other subunits of the ATAC complex and binds to H3K27ac via its YEATS domain (*Mi et al., 2017*). The ZZ domain of ZZZ3 binds the histone H3 tail and in combination with YEATS2 anchors the ATAC complex to acetylated H3 (*Mi et al., 2018*). These proteins bind to other proteins in the complex including Ada2a as well as the GCN5 catalytic subunit (*Mi et al., 2017*; *Mi et al., 2018*; *Grant et al., 1997*). GCN5 or the closely related PCAF subunit (*Nagy and Tora, 2007*) are responsible for acetylating H3K9, which subsequently opens the surrounding chromatin structure promoting transcription (*Mi et al., 2017*). In addition, KAT14, a third subunit of the ATAC complex, was identified by BioID mass spectrometry to interact with TAZ-CAMTA1 and YAP-TFE3 (*Figure 6C*), further emphasizing the functional significance of this complex and an additional impetus to prioritize YEATS2 and ZZZ3 for further study. Knock-down of *YEATS2* in SW872 TAZ-CAMTA1 cells was confirmed by quantitative RT-PCR (*Figure 8—figure supplement 1A*) and resulted in loss of anchorage-independent growth by soft agar assay (*Figure 8—figure supplement 1B*). Similarly, knock-down of ZZZ3 in SW872 TAZ-CAMTA1 cells was confirmed by western blot (*Figure 8—figure supplement 1C*) and the loss of anchorage-independent phenotype was replicated by soft agar assay (*Figure 8—figure supplement 1D*). We then performed co-immunoprecipitation studies to validate the BioID experiments. Of the three subunits of the ATAC complex identified by BioID mass spectrometry, KAT14 showed the strongest interaction with TAZ-CAMTA1 and YAP-TFE3 by co-immunoprecipitation (*Figure 8—figure supplement 1E–G*).

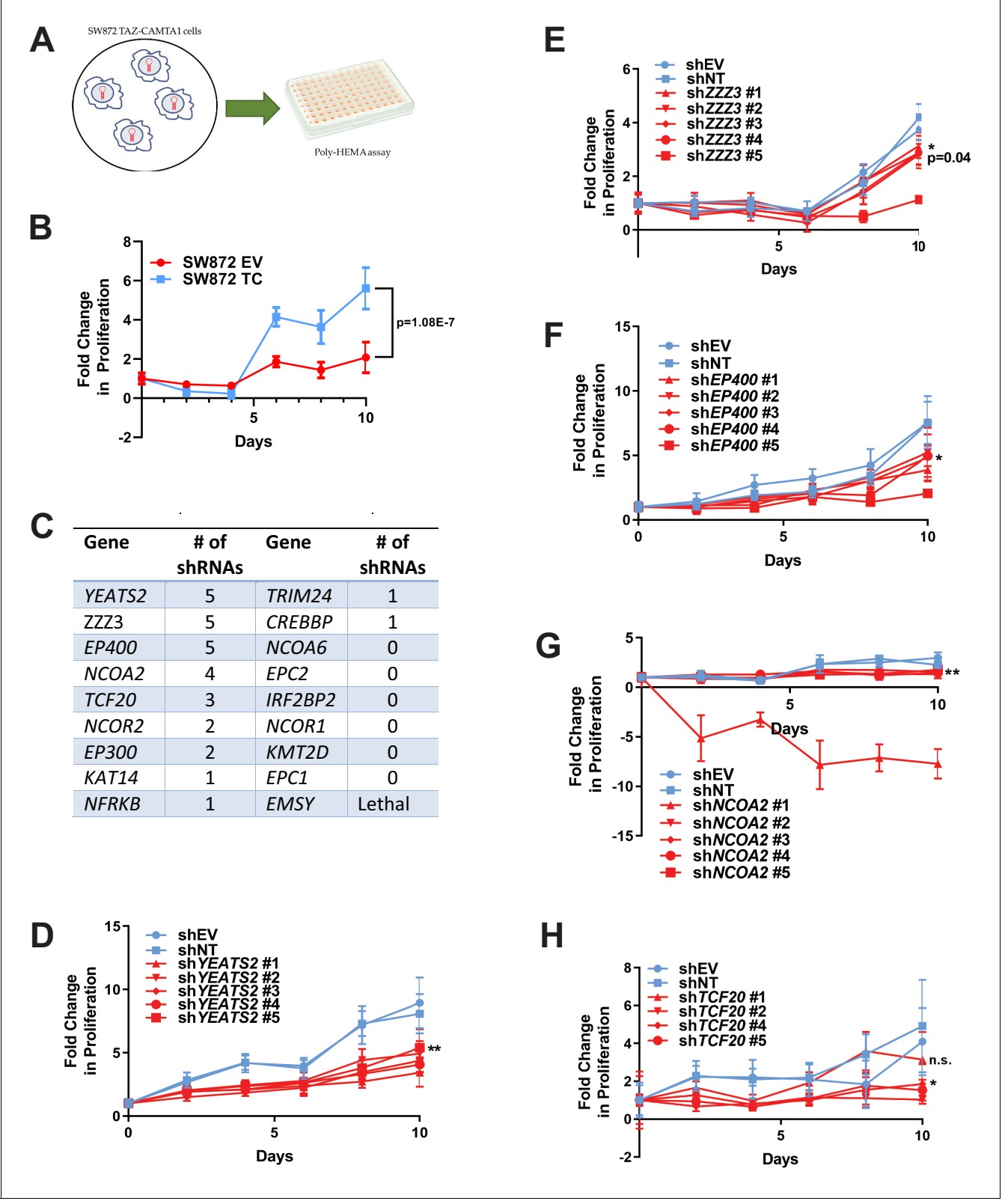

**Figure 7.** RNAi screen shows subunits of the ATAC complex mediate the transforming capacity of TAZ-CAMTA1 and YAP-TFE3. (A) Overview of RNAi screen. SW872 cells expressing TAZ-CAMTA1 are transduced with five shRNAs for each gene in the shared TAZ-CAMTA1/YAP-TFE3 interactome and proliferation assayed on poly-HEMA-coated plates. (B) Poly-HEMA assay with SW872 cells expressing TAZ-CAMTA1. (C) Table with rank ordering of shared prey proteins after RNAi screen according to shRNAs causing a decrease in anchorage-independent growth. (D–H) Proliferation curves for prey

*Figure 7 continued on next page*

*Figure 7 continued*

proteins/genes with the highest number of shRNAs demonstrating a reduction in anchorage-independent growth on poly-HEMA coated plates. For poly-HEMA proliferation assays, statistical significance was evaluated using fold change increase in proliferation at day 10 with an unpaired two-tailed *t*-test. Error bars were used to define one standard deviation. For all panels, \*\*\*\*$p<0.0001$, \*\*\*$p<0.001$, \*\*$p<0.01$, \*$p<0.05$.

The online version of this article includes the following source data and figure supplement(s) for figure 7:

**Source data 1.** Oligonucleotide sequences for shRNA constructs used in the RNAi screen.
**Figure supplement 1.** RNAi screen identifies epigenetic modifiers important for the activity of TAZ-CAMTA1 and YAP-TFE3.

To investigate the role of the ATAC complex in mediating the transcriptional programs of TAZ-CAMTA1 and YAP-TFE3, we performed RNA-Seq on SW872 cells expressing TAZ-CAMTA1 and YAP-TFE3 with stable knock-down of *YEATS2* or *ZZZ3*. Principal component analysis showed that knock-down of *YEATS2* and *ZZZ3* altered both the TAZ-CAMTA1 and YAP-TFE3 transcriptomes, compared to shRNA control (*Figure 8A*). Knock-down of YEATS2 and ZZZ3 had a greater effect on the final transcriptional program of YAP-TFE3 than TAZ-CAMTA1, likely reflecting the smaller number of epigenetic modifiers interacting with YAP-TFE3 as compared to TAZ-CAMTA1. Aligning the most down-regulated genes for TAZ-CAMTA1 sh*YEATS2*, TAZ-CAMTA1 sh*ZZZ3*, YAP-TFE3 sh*YEATS2*, and YAP-TFE3 sh*ZZZ3* revealed an enrichment of extracellular matrix proteins (*FBLN5*, *SPP1*, *GPC4*, and *LAMA1*) (*Figure 8B* and *Figure 8—source data 1*) and validated by qRT-PCR (*Figure 8C*). iPathwayGuide analysis showed that MAPK signaling, a pathway significantly upregulated in SW872 YAP-TFE3 cells (*Figure 3I*) was significantly down-regulated with *YEATS2* knock-down (*Figure 8D* and *Figure 8—figure supplement 2A*). Further, PI3K-Akt signaling, activated in SW872 cells expressing TAZ-CAMTA1 or YAP-TFE3 (*Figure 3H,I* and *Figure 8—figure supplement 2B*), was also decreased in SW872 cells expressing YAP-TFE3 and either *YEATS2* or *ZZZ3* shRNA (*Figure 8D* and *Figure 8—figure supplement 2C–D*). Since the ATAC complex is a HAT complex thought to increase the amount of transcriptionally available chromatin, we intersected genes differentially expressed with *YEATS2* or *ZZZ3* knockdown with open chromatin unique to SW872 YAP-TFE3 expressing cells. We found that 71% of differentially expressed genes (DEG) with *YEATS2* knock-down and 74% of DEG with *ZZZ3* knock-down were within euchromatin unique to YAP-TFE3 expressing cells (*Figure 8E,F*). These findings support a working model that the ATAC complex interacts with the C termini of TAZ-CAMTA1 and YAP-TFE3 via the KAT14 subunit at predominantly H3K27ac-positive active enhancer regions, and subsequently acetylates H3K9 to modulate a TEAD-based transcriptional program (*Figure 8G*).

Towards validating these findings in clinical samples, we noted that ECM proteins were consistently represented in previous analysis of the TAZ-CAMTA1 and YAP-TFE3 transcriptomes includes those altered by knock-down of components of the ATAC complex (*Figure 3H* and *Figure 8B*). To validate expression of various ECM proteins upregulated in TAZ-CAMTA1 and YAP-TFE3 expressing cells (*Figure 3—source data 2*), we performed immunohistochemistry for COL1A1 (Collagen type I alpha one chain), CTGF (Connective tissue growth factor), VTN (vitronectin), and FBLN5 (Fibulin-5) on EHE clinical samples and compared expression with two histological mimics of EHE, epithelioid angiosarcoma (E-AS) and epithelioid hemangioma (EH). While CTGF, FBLN5, and COL1A1 were expressed in EHE (*Figure 8—figure supplement 2E and F*), they were also expressed in E-AS and EH. In contrast, VTN was expressed in all EHE samples, but was essentially absent in epithelioid angiosarcoma and epithelioid hemangioma (*Figure 8H*).

To determine if expression of *YEATS2* and *ZZZ3* could be prognostically important in sarcomas other than EHE, we utilized RNA-Seq expression data from sarcoma clinical samples from The Cancer Genome Atlas (TCGA) database. Via Kaplan-Meier analysis, we showed that increased expression of *YEATS2* (*Figure 8I*) or *ZZZ3* (*Figure 8J*) predicted a poorer prognosis across different histological types of sarcoma, suggesting the ATAC complex may be a key oncogenic driver in multiple sarcomas in addition to EHE. This is consistent with recent findings identifying amplification of *YEATS2* in sarcomas such as well differentiated liposarcoma and dedifferentiated liposarcoma (*Beird et al., 2018*).

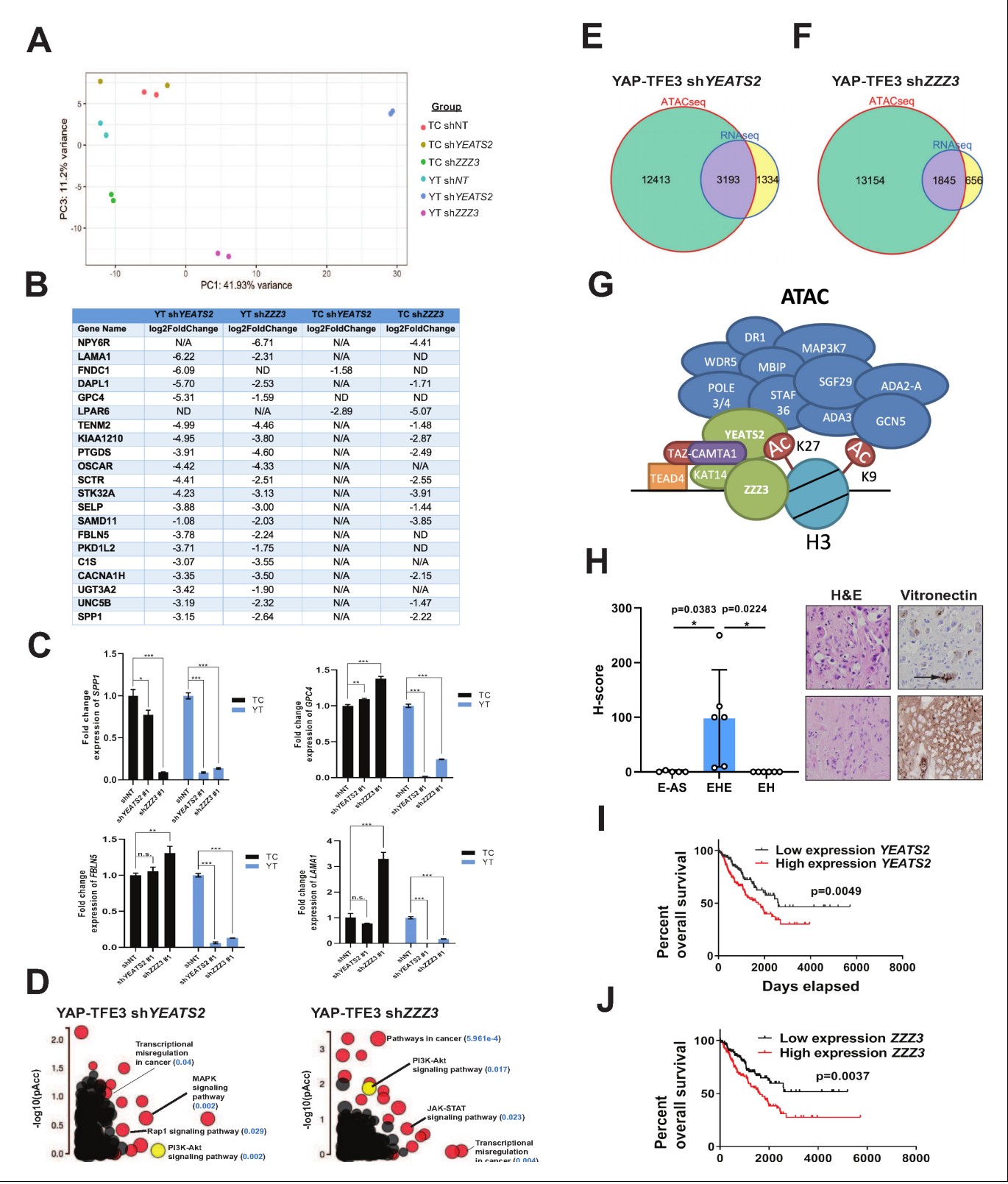

**Figure 8.** The ATAC complex is critical for the oncogenic transcriptomes of the fusion proteins. (A) Principal component analysis showing the effect of knock-down of *YEATS2* or *ZZZ3* on the TAZ-CAMTA1 and YAP-TFE3 transcriptomes. (B) Genes with at least a two-fold decrease in at least two of the four conditions (YT sh*YEATS2*, YT sh*ZZZ3*, TC sh*YEATS2*, and TC shZZZ3); top genes most down-regulated according to aforementioned filter. N/A indicates gene was not differentially expressed in the data set. ND indicates there was no decrease in expression. (C) Extracellular matrix-related

*Figure 8 continued on next page*

*Figure 8 continued*

downregulated genes in part **B** validated by quantitative RT-PCR. (**D**) Scatter plot of two types of pathway enrichment evidence: probability of over-representation (pORA) and probability of accumulation (pAcc) as calculated by iPathwayGuide for SW872 YAP-TFE3 cells with knock-down of *YEATS2* or *ZZZ3*. (**E**) Intersection of differentially expressed genes after knock-down of *YEATS2* in SW872 YAP-TFE3 cells with open chromatin regions unique to YAP-TFE3 expressing cells (not present with empty vector). (**F**) Intersection of differentially expressed genes after knock-down of *ZZZ3* in SW872 YAP-TFE3 cells with open chromatin regions unique to YAP-TFE3 expressing cells (not present with empty vector). (**G**) Working model of TAZ-CAMTA1: ATAC complex. (**H**) Vitronectin expression in epithelioid hemangioendothelioma (EHE) compared to epithelioid angiosarcoma (E–AS) and epithelioid hemangioma (EH). Vitronectin expression could be more focal (top example of EHE) and secreted in the ECM between neoplastic cells (demonstrated by arrow) or more diffusely expressed in the ECM (bottom example of EHE). Overall survival curves as a function of *YEATS2* (**I**) and *ZZZ3* RNA expression (**J**) across different histological types of sarcoma utilizing RNA-Seq expression data for The Cancer Genome Atlas (TCGA). RNA-Seq experiments in SW872 cells were performed using biological duplicates for each of the conditions. For gene expression data, the population was set to the total number of recovered genes with the mean of five counts across all samples. ATAC-Seq experiments in SW872 cells were performed using biological triplicates for each of the conditions (expression constructs). For ATAC-Seq analysis, the population was set as the total number of genes annotated across all conditions. For quantitative RT-PCR, standard deviation was calculated from fold change values for each triplicate. Error bars were used to define one standard deviation. For Kaplan-Meier curves significance was determined by Log-rank (Mantel-Cox) test. Each experiment was repeated at least twice. For all panels, ****p<0.0001, ***p<0.001, **p<0.01, *p<0.05.

The online version of this article includes the following source data and figure supplement(s) for figure 8:

**Source data 1.** Differentially expressed genes in SW872 cells expressing TAZ-CAMTA1 or YAP-TFE3 after *YEATS2* or *ZZZ3* knock-down.
**Figure supplement 1.** Validation of mass spectrometry/RNAi screen.
**Figure supplement 2.** The ATAC complex modulates expression of the MAPK signaling pathway, the PI3K signaling pathway, and extracellular matrix (ECM) proteins.

## Discussion

Here, we provide new insights into the endothelial cancer EHE by studying the disease defining fusion proteins TAZ-CAMTA1 and YAP-TFE3. We find that each fusion protein drives a related but distinct transcriptional program, which is underpinned by the fact that they both engage the genome at least in part via the TEAD transcription factors, as do wild-type YAP and TAZ. Mechanistically, using a paired proteomic and genomic screen, we identify the ATAC complex as a key epigenetic mediator enabling both TAZ-CAMTA1 and YAP-TFE3 to modulate the chromatin environment and transcription, thus transforming cells. This is despite the fact that the amino termini of TAZ and YAP are fused to distinct transcription factors, CAMTA1 and TFE3, respectively.

### TAZ-CAMTA1 and YAP-TFE3 drive a predominantly TEAD-based transcriptional program

Our current studies show that rather than simply recapitulating the TAZ and YAP transcriptional programs, TAZ-CAMTA1 and YAP-TFE3 drive a transcriptome that overlaps with, but is significantly different from that of each full-length protein. Further, the degree to which the fusion protein transcriptomes do overlap with the TAZ/YAP transcriptional programs is due to shared TEAD transcription factor binding. This is borne out via evaluation of the chromatin binding profiles of the two fusion proteins which showed similar levels of enrichment for TEAD-binding sites compared to full-length TAZ and YAP. Our findings are in keeping with previous functional studies of TAZ-CAMTA1 showing that it requires binding to TEAD4 in order to drive its oncogenic transcriptional program and transform cells (*Tanas et al., 2016*).

### The fusion protein transcriptomes differ from the full-length TAZ/YAP transcriptomes due to altered DNA binding

Although the transcriptional programs of TAZ-CAMTA1 and YAP-TFE3 significantly overlap with the TAZ/YAP transcriptomes, they also differ in important ways. In both cell lines examined here, approximately 20–40% of differentially expressed genes were unique to the TAZ-CAMTA1- and YAP-TFE3-induced transcriptomes, indicating that the fusion proteins do not only regulate transcription via TEADs. Accordingly, unbiased motif enrichment analysis of TAZ-CAMTA1 revealed an enrichment for the EGR2 binding motif while YAP-TFE3 showed an enrichment for the MITF-binding motif. This indicates that the genomic occupancy of TAZ-CAMTA1 and YAP-TFE3 is mediated by the C termini of CAMTA1 and TFE3 in addition to interactions with the TEAD transcription factors. Bioinformatic approaches predicting the chromatin binding of transcription factors that contain multiple

DNA binding sites have been developed (*Vandel et al., 2019*), which in combination with structure-function studies should yield additional insight into how these chimeric transcription factors interface with the genome.

## The fusion protein transcriptomes differ from the TAZ/YAP transcriptomes due to shifts in the chromatin landscape

We found that TAZ-CAMTA1 and YAP-TFE3 increased the amount of transcriptionally active chromatin when compared to TAZ and YAP overexpression. Although genes embedded in transcriptionally accessible portions of the genome showed a degree of overlap between the fusion proteins and either TAZ or YAP, hypergeometric analysis showed that the euchromatin landscape induced by TAZ-CAMTA1 more closely resembled that of CAMTA1, while that of YAP-TFE3 showed an almost equivalent level of similarity to TFE3 as it did to YAP. This observation is reflected in the PCA analysis, which showed that the distribution of transposase accessible chromatin for the fusion proteins was more closely related to each other, CAMTA1, and TFE3 than to TAZ/YAP. This indicates that CAMTA1 and TFE3 confer chromatin remodeling properties to the fusion proteins, while the N termini of TAZ and YAP essentially contribute the TEAD binding domain. This is consistent with previous studies that have demonstrated that TAZ/YAP's interactions with epigenetic modifiers are key to their function. For example, in *Drosophila*, Yorkie (YAP orthologue) can interact with Ncoa6, a subunit of the Trithorax-related histone H3 lysine four methyltransferase complex to drive tissue growth. Importantly, conjugation of Ncoa6 to Scalloped (TEAD orthologue) alone was sufficient to functionally activate the Yorkie transcriptional program (*Oh et al., 2014*; *Qing et al., 2014*). In mammalian/cancer cells, YAP/TAZ-driven transcription is facilitated by binding to BRD4, a member of the bromodomain and extraterminal (BET) protein family of epigenetic modifiers that mediates interaction between enhancer sequences and associated promoters (*Zanconato et al., 2018*). Taken together, the above studies indicate that additional fine tuning of the TEAD-based transcriptional program is essential to TAZ/YAP function in both physiological and cancer-related contexts.

Although the overlap in euchromatin profile between the fusion proteins and CAMTA1/TFE3 can be partially explained by DNA-binding elements conferred by the C termini of CAMTA1 and TFE3, it also raises questions about how the TAZ-CAMTA1:ATAC and YAP-TFE3:ATAC complexes form. It remains to be determined how the fusion proteins dynamically recruit epigenetic modifiers to contact points in the genome where TEAD and non-TEAD binding sites (e.g. MITF or EGR2) are in close proximity.

## The enhanced euchromatin landscape in cells expressing TAZ-CAMTA1 and YAP-TFE3 is mediated by the ATAC complex

Although we found that TAZ-CAMTA1 and YAP-TFE3 interact with numerous epigenetic complexes, the combined proteomic/RNAi screen demonstrated that the ATAC complex is the most functionally relevant for cell transformation. Ada2a-containing acetyltransferase (ATAC) and Spt-Ada-Gcn5-acetyltransferase (SAGA) are two closely related metazoan histone acetyltransferase complexes that both contain GCN5 or the closely related PCAF histone acetyltransferase catalytic subunits. While the SAGA complex is predominantly localized to promoter sequences, the ATAC complex is found at both promoter and enhancer sequences (*Krebs et al., 2011*). This is consistent with our observation that TAZ-CAMTA1 and YAP-TFE3, which are predominantly localized at enhancer sequences, interact primarily with the ATAC complex rather than the SAGA complex. Although the ATAC complex can regulate both housekeeping and tissue specific functions, its binding to enhancer sequences is cell-type specific (*Krebs et al., 2011*). Furthermore, knock-down of subunits of the ATAC complex in cells expressing TAZ-CAMTA1 or YAP-TFE3 affected only a limited set of active genes rather than globally altering gene expression, another key piece of data demonstrating that the ATAC complex modulates a specific transcriptional program. The implication of these data is that the ATAC complex is recruited to a particular set of target genes and interacts with a specific set of transcription factors rather than ubiquitously interacting with many different transcription factors across the genome. This is corroborated by our BioID mass spectrometry data which showed that the ATAC complex preferentially binds to both fusion proteins as compared to full-length TAZ and YAP.

Importantly, our study also indicates that the ATAC complex represents a potential therapeutic target for EHE, a sarcoma currently lacking an effective medical therapy, regardless of which fusion protein it harbors. In addition, RNA expression profiling data show that expression of *YEATS2* and *ZZZ3* have prognostic significance in sarcoma clinical samples across various histological types, suggesting that the ATAC complex might represent a key oncogenic driver in other sarcomas. Future studies elucidating therapeutic strategies targeting the ATAC complex are warranted.

## TAZ-CAMTA1 and YAP-TFE3 have convergent oncogenic properties

Despite the observation that CAMTA1 and TFE3 are from entirely different protein families, several lines of evidence reveal that the function of TAZ-CAMTA1 and YAP-TFE3 are remarkably similar. As mentioned above, the two fusion proteins are similar in terms of the genes they induce, their chromatin-binding profiles, and how they alter the chromatin landscape. This raises the question of whether other TAZ/YAP gene fusions show similar convergence of function. Instead of point mutations, gene fusions are the most common genetic alterations of TAZ(*WWTR1*) and YAP in cancer. Structurally similar *WWTR1*(TAZ)/*YAP1* gene fusions have also been identified in nasopharyngeal carcinoma (*YAP1-MAMLD2*) (*Valouev et al., 2014*), lung cancer (*YAP1-BIRC2; WWTR1-SLC9A9*) (*Dhanasekaran et al., 2014*), cervical squamous cell carcinoma and endocervical adenocarcinoma (*YAP1-SS18*) (*Hu et al., 2018*; *Szulzewsky et al., 2020*), poromas/porocarcinomas (*YAP1-MAML2* and *YAP1-NUTM1*) (*Sekine et al., 2019*), ependymomas (*YAP1-MAMLD1* and *YAP1-FAM118B*) (*Szulzewsky et al., 2020*; *Pajtler et al., 2015*; *Pajtler et al., 2019*), ossifying fibromyxoid tumor (*KDM2A-WWTR1*) (*Kao et al., 2017*), *NF2*-wildtype meningiomas (*YAP1-MAML2, YAP1-PYGO1, YAP1-LMO1*) (*Sievers et al., 2020*), retiform and composite hemangioendotheliomas (*YAP1-MAML2*) (*Antonescu et al., 2020*), and sclerosing epithelioid fibrosarcoma (*YAP1-KMT2A*) (*Puls et al., 2020*; *Kao et al., 2020*). Here, we have outlined an approach integrating NGS modalities with combined proteomic/genetic screens to gain insight into the transcriptional regulatory mechanisms of the TAZ-CAMTA1 and YAP-TFE3 fusion proteins. At its most essential, the mechanism of the two fusion proteins is the same in that a TEAD-based transcriptional program is modulated by interaction with an epigenetic modifier, the ATAC complex. Additional studies are needed to determine if YAP/TAZ/TEAD fusion proteins in other cancers also drive tumorigenesis via the ATAC complex or do so by recruiting different chromatin-modifying complexes.

# Materials and methods

**Key resources table**

| Reagent type (species) or resource | Designation | Source or reference | Identifiers | Additional information |
|---|---|---|---|---|
| Cell line (*Mus musculus*) | NIH 3T3 cells | ATCC | CRL-1658; RRID:CVCL_0594 | *Mycoplasma* negative |
| Cell line (*Homo sapiens*) | SW872 cells | ATCC | HTB92; RRID:CVCL_1730 | *Mycoplasma* negative |
| Transfected construct (*H. sapiens*) | pPGS-3HA-TEAD1 | Dr. Kunliang Guan (Addgene) | #33055; RRID:Addgene_33055 | |
| Transfected construct (*H. sapiens*) | 8xGTIIC-luciferase | Dr. Stefano Piccolo (Addgene) | #34615; RRID:Addgene_34615 | |
| Transfected construct (*H. sapiens*) | pRP-CMV-3xMyc-KAT14 | Vector builder | | |
| Commercial assay or kit | SimpleChIP Enzymatic Chromatin IP Kit | Cell Signaling | #9003 | |
| Antibody | Anti-DYKDDDDK antibody | Cell Signaling | #14793; RRID:AB_2572291 | ChIP grade antibody |

## Expression constructs

Double Flag TAZ-CAMTA1 and Double Flag CAMTA1 were cloned into pBabeNeo as previously described. Double Flag TAZ4SA was derived utilizing previously constructed TAZ S89A (*Tanas et al., 2016*). Additional S66A, S117A, and S311A mutations were introduced by site directed mutagenesis using the QuikChange II Site-directed mutagenesis kit (Agilent #200521) and the following primers:

TAZ S66A Primer – Forward
5' – ctcgcgccagtccgccaccgactcgtcg – 3'
TAZ S66A Primer – Reverse
5' – cgacgagtcggtggcggactggcgcgag – 3'
TAZ S117A Primer – Forward
5' – cctccgccagcaggcctacgacgtgaccgac – 3'
TAZ S117A Primer – Reverse
5' – gtcggtcacgtcgtaggcctgctggcggagg – 3'
TAZ S311A Primer – Forward
5' – cattcgagggagcaggccactgacagtggcct – 3'
TAZ S311A Primer – Reverse
5' – aggccactgtcagtggcctgctccctcgaatg – 3'

Double Flag YAP, YAP-TFE3 and TFE3 constructs were cloned into pBabeNeo as previously described (*Tanas et al., 2016*).

YAP S61A Primer – Forward
5' – TCCGCGGGGACGCGGAGACCGAC – 3'
YAP S61A Primer – Reverse
5' – GTCGGTCTCCGCGTCCCCGCGGA – 3'
YAP S109A Primer – Forward
5' – CACTCCCGACAGGCCGCTACTGATGCAGGCAC – 3'
YAP S109A Primer – Reverse
5' – GTGCCTGCATCAGTAGCGGCCTGTCGGGAGTG – 3'
YAP S127A Primer – Forward
5' – AGCATGTTCGAGCTCATGCCTCTCCAGCTTCT – 3'
YAP S127A Primer – Reverse
5' – AGAAGCTGGAGAGGCATGAGCTCGAACATGCT – 3'
YAP S164A Primer – Forward
5' – CAGCATCTTCGACAGTCTGCTTTTGAGATACCTGATG – 3'
YAP S164A Primer – Reverse
5' – CATCAGGTATCTCAAAAGCAGACTGTCGAAGATGCTG – 3'
YAP S397A Primer – Forward
5' – CTATCACTCTCGAGATGAGGCTACAGACAGTGGACTAAGC – 3'
YAP S397A Primer – Reverse
5' – GCTTAGTCCACTGTCTGTAGCCTCATCTCGAGAGTGATAG – 3'

The pPGS-3HA-TEAD1 construct was obtained courtesy of Dr. Kunliang Guan.

Triple Myc-tagged KAT14 plasmid (Vector ID: VB18 p18-119pmf; Vector name: pRP[Exp]-Puro-CMV>BamHI/3xMyc-KAT14/EcorRI) was synthesized by Vector builder, Shenandoah, TX.

## Cell culture

NIH 3T3 mouse fibroblasts and SW872 human liposarcoma cells were obtained from American Type Culture Collection (ATCC, Manassas, VA, USA). NIH 3T3 cells were cultured in DMEM media containing 10% bovine calf serum (Invitrogen-Life Technologies), 1 mM sodium pyruvate, and 50 μg/mL penicillin/streptomycin. SW872 cells were cultured in DMEM media (according to ATCC recommendations) containing 10% fetal bovine serum (Invitrogen-Life Technologies), 1 mM sodium pyruvate, and 50 μg/mL penicillin/streptomycin. All cells were cultured at 37°C and 5% $CO_2$.

## Transfection and retroviral transduction

Retroviral transfection with pBabeNeo constructs was performed by transfecting PhoenixA retroviral packaging cells with 10 μg of plasmid DNA. Transfection was done using Lipofectamine Reagent and Plus Reagent (Invitrogen-Life Technologies) according to manufacturer's instructions.

Supernatant was collected at 48 and 72 hr after transfection, filtered with 0.45 µm filters and supplemented with 8 µg/mL polybrene (EMD Millipore, Burlington, MA, USA). Serial transductions (48 and 72 hr supernatants) were applied to either SW872 or NIH 3T3 cells for 8 hr each. Pooled stable lines were generated by selecting with G418 for 2 weeks.

## RNA interference-mediated silencing

PLKO.1-puro constructs for knock-down of Tead1 were obtained from Sigma-Aldrich (MISSION shRNAs). Constructs were transfected into LentiX (HEK293T) cells, along with pcMVΔ8.12 and pVSVG packaging plasmids, using Lipofectamine Reagent and Plus Reagent (Invitrogen) according to manufacturer's instructions. Supernatants were collected at 48 hr, filtered using 0.45 µm filters, and polyethylene glycol (PEG) 8000 was added to a final concentration of 12%, and stored at 4°C overnight. The following day, 48 hr supernatants were centrifuged and viral pellets were resuspended in 0.45 µm filtered 72 hr media and supplemented with 8 µg/mL (EMD Millipore, Burlington, MA, USA) added to either SW872 or NIH 3T3 cells overnight. Pooled stable lines were generated by selection in puromycin.

## Antibodies for western blot

Anti-FLAG antibody (mouse monoclonal clone M2 (catalog # F3165; RRID:AB_259529) utilized for immunofluorescence and western blot (1:1000) was obtained from Sigma-Aldrich (St. Louis, MO, USA). Anti-β-actin antibody (AC-15; catalog #A544; RRID:AB_4767441) utilized for western blot (1:10,000) was obtained from Sigma-Aldrich (St. Louis, MO, USA). Alexa 568-conjugated secondary antibody (catalog# A11031; RRID:AB_144696) was obtained from Invitrogen-Life Technologies (Grand Island, NY, USA). Anti-alpha tubulin antibody (clone DM1A, catalogue #T9026; RRID:AB_477593) was obtained from Sigma-Aldrich (St. Louis, MO, USA). Anti-H3 antibody (clone 1G1, catalogue# sc-517576; RRID:AB_2848194) was obtained from Santa Cruz Biotechnology (Dallas, TX, USA). Anti-TEAD1 antibody (clone EPR3967 *Errani et al., 2011*, catalogue #AB133533; RRID:AB_2737294) was obtained from Abcam (Cambridge, MA, USA). Anti-HA antibody (clone F-7, sc-7392; RRID:AB_627809) was obtained from Santa Cruz Biotechnology (Dallas TX, USA). Anti-ZZZ3 antibody (catalogue# ab118800; RRID:AB_10901749) was obtained from Abcam (Cambridge, MA, USA). Anti-Myc antibody (catalogue # 2272S; RRID:AB_10692100) was obtained from Cell Signaling Technologies (Danvers, MA, USA). Anti-TFE3 antibody (catalogue# HPA023881; RRID:AB_1857931) was obtained from Sigma-Aldrich. Horseradish peroxidase-conjugated secondary antibodies used for western blots were obtained from Santa Cruz Biotechnology (catalog# sc-2055; RRID:AB_631738, sc-2054; RRID:AB_631748, or sc-2033; RRID:AB_631729) and used at 1:5000 or 1:10,000).

## Immunohistochemistry

Antibodies used for immunohistochemistry include anti-Ki-67 from Abcam (ab#16667, clone SP6). Anti-vitronectin antibody (mouse anti-human monoclonal antibody, catalogue# LS-B2118), anti-COL1A1 antibody (clone 3G3, catalogue# LS-B5932), anti-CTGF antibody (rabbit polyclonal, catalogue# LS-B3284), and anti-fibulin five antibody (rabbit polyclonal, catalogue# LS-B14518) obtained from LifeSpan BioSciences (Seattle, WA). Dilution for the above antibodies were as follows: anti-Ki-67 (1:100), anti-vitronectin (1:400), anti-COL1A1 (1:200), anti-CTGF (1:200), and anti-fibulin 5 (1:200). For all of those markers, the antigen retrieval was Citrate Buffer pH 6.0, 110°C for 15 min, in Biocare Decloaker (Biocare Medical, Concord, CA, USA). Secondary antibodies were anti-mouse for VTN and COL1A1, anti-rabbit for Ki-67, FBLN and CTGF, from Dako Envision HRP System (Dako North America, Inc, Carpentaria, CA, USA).

## Western blot

Cell pellets were lysed in radioimmunoprecipitation assay (RIPA) buffer, containing cOmplete Protease Inhibitor Cocktail (EDTA-free) (Roche) and PhosSTOP Phosphatase Inhibitor Cocktail (Roche) according to the manufacturer's instructions. Total protein concentration was measured using Pierce BCA Protein Assay Kit (ThermoFisher Scientific, Waltham, MA, USA). Between 50 and 100 µg of total protein was loaded onto a gradient (4–15%) polyacrylamide gel (BioRad, Hercules, CA, USA). Proteins were then transferred to a polyvinylidene difluoride (PVDF) membrane and probed with antibodies described above. Each experiment was repeated at least twice.

## Co-immunoprecipitation

One mg of total cellular protein (whole cell lysates) were cleared by incubating with 2.0 µg of the control IgG and 20 µL vortexed Protein G PLUS-Agarose beads (Santa Cruz, sc-2002) for 30 min with rocking at 4°C. Supernatant transferred and incubated with 10 µg primary antibody or IgG control and incubated at 4°C overnight with rocking. 40 µL of vortexed Protein G PLUS agarose beads added and incubated at 4°C on a rocker platform for 4 hr. After centrifugation and removal of the supernatant, the cell pellet was re-suspended and washed three times with 1 mL cold 1X PBS followed by an additional wash with 1 mL 1X RIPA buffer. The pellet/beads were then resuspended in 2X Laemmli buffer, boiled, then loaded onto polyacrylamide gel.

## Nuclear and cytoplasmic fractionation

Nuclear and cytoplasmic fractionation was performed with the Nuclear Extract Kit (Active Motif, 40010). Cells collected from 15 cm cell culture plates were centrifuged and resuspended in 500 µL 1X Hypotonic buffer and incubated for 5 min on ice. Twenty-five µL of detergent was added to resuspended cells which were subsequently vortexed for 10 s. After centrifugation, the supernatant cytoplasmic fraction was removed and stored. Remaining nuclear pellet was resuspended in 50 µL complete lysis buffer followed by vortexing. After centrifugation, the supernatant nuclear fraction was removed and stored.

## Immunofluorescence staining

Cells were plated in chambered slides ($2.5 \times 10^4$ for sparse conditions or $5 \times 10^4$ for confluent conditions) and allowed to grow until desired timepoint. Cells were fixed with 4% paraformaldehyde (in 1X PBS) for 15 min and then washed with 1X PBS. Cells were permeabilized and blocked with 0.3% Triton X-100% and 3% fetal bovine serum for 30 min. Primary antibodies, as described above, were diluted in 3% fetal bovine serum and incubated on the cells overnight at 4°C in a humidity chamber. The following morning, the primary antibody was removed, cells were washed with 1X PBS, and then incubated with Alexa Fluor 568-conjugated secondary antibody (described above) for 45 min to 1 hr at room temperature. Immunofluorescence was visualized using a Leica DFC3000G microscope/ camera and Leica Application Suite Advanced Fluorescence imaging software or an Olympus BX-61 microscope (Tokyo, Japan) with cellSens imaging software. Each experiment was repeated at least twice.

## Soft agar assay

The base layer of 0.5% agarose was plated into 6-well plates (2 mLs/well) and allowed to solidify. For SW872 cells, $5 \times 10^3$ cells/2 mL in 0.35% agarose was added to form the top layer. For NIH 3T3 cells, $1 \times 10^4$ cells/2 mL in 0.35% agarose was added to form the top layer. Plates were left in the hood and allowed to solidify at room temperature for 3 hr. Colonies were allowed to grow for 2–3 weeks at 37°C and 5% $CO_2$ before imaging. Each experiment was repeated at least twice.

## Poly-HEMA assay

Poly(2-hydroxyethyl methacrylate), or poly-HEMA, solution was made at 20 mg/mL in 95% ethanol. 96-well tissue culture plates were coated with 130 µL of poly-HEMA solution and UV sterilized overnight. Cells were plated ($2 \times 10^3$ cells/100 µL for SW872 or $3 \times 10^3$ cells/100 µL for NIH 3T3) into multiple wells and incubated at 37°C and 5% $CO_2$. Proliferation was assessed every other day by adding 10 µL of Cell Counting Kit-8 reagent (Dojindo Molecular Technologies, Inc, Rockville, MD, USA). The plates were then incubated for 1 hr at 37°C and 5% $CO_2$ and then absorbance at 450 nm was measured using a Synergy H1 Hybrid Multi-Mode Microplate Reader (Biotek, Winooski, VT, USA).

## Xenografts

NOD scid gamma (NSG) mice were obtained from The Jackson Laboratory (Bar Harbor, ME, USA) and carry the strain NOD.Cg-Prkdcscid Il2rgtm1Wjl/SzJ (RRID:IMSR_JAX:005557). All animal work were approved by the University of Iowa Institutional Animal Use and Care Committee. SW872 cells containing either empty vector, YAP-TFE3, or TAZ-CAMTA1 and NIH 3T3 cells containing empty vector, YAP-TFE3, or TAZ-CAMTA1 ($1 \times 10^7$ cells/500 µL PBS) were injected into the flank of NSG

mice. Mice were 6 months of age at the time of NIH 3T3 injection, and 2–3 months of age at the time of SW872 injection. Xenografts were repeated twice, using 5–10 mice per group.

## Quantitative reverse transcription polymerase chain reaction

SW872 and NIH 3T3 cells were scraped and collected using TRIzol reagent (Ambion-Life Technologies). Total RNA was isolated using the PureLink RNA mini kit (Invitrogen-ThermoFisher Scientific). On-column DNase (Invitrogen) treatment was performed according to manufacturer's instructions. Purified RNA was quantified using a NanoDrop (ThermoFisher) and 1 µg was converted to cDNA using SuperScript III First-Strand Synthesis System (Invitrogen) according to manufacturer's instructions. PCR amplification was performed in technical triplicates on the Applied Biosystems QuantStudio 3 Real-Time PCR System (Applied Biosystems-ThermoFisher). TaqMan Universal PCR Master Mix (Applied Biosystems-ThermoFisher) and PrimeTime standard qPCR primer/probe sets from Integrated DNA Technologies (Iowa City, IA, USA) were utilized. The qPCR cycling conditions were as follows: $95\ ^{\circ}C^{10:00}(95\ ^{\circ}C^{0:15},\ 60\ ^{\circ}C^{1:00})_{40}$. Relative quantitation was performed utilizing the delta-delta $C_T$ method and *POLR2A/Polr2a* (RNA Polymerase II) $C_T$ values as the reference control. Each experiment was repeated at least twice. The following primers and probes were used:

### Human

*CCN2*-F 5'-ACCAATGACAACGCCTCC-3'
*CCN2*-R 5'-TTGGAGATTTTGGGAGTACGG-3'
*CCN2*-Probe 5'-TGCGAAGCTGACCTGGAAGAGAAC-3'
*CCN1*-F 5'-GGGATTTCTTGGTCTTGCTG-3'
*CCN1*-R 5'- CCAATGACAACCCTGAGTGC-3'
*CCN1*-Probe 5'-TTCTTGCCCTTTTTCAGGCTGCTG-3'
*AJUBA*-F 5'-GCCTCTACCACACCCAGT-3'
*AJUBA*-R 5'-ACAGTACACAGAGCCATTGAC −3'
*AJUBA*-Probe 5'-/56-FAM/TGCAACGCA/ZEN/AAGTTCGCCCA/3IABkFQ/−3'
*FBLN5*-F 5'-CCTCTTATATGCCGCTTTGGA-3'
*FBLN5*-R 5'-CCCGCCTTCAGTATTGATGC-3'
*FBLN5*-Probe 5'-/56-FAM/TCTGTTGCA/ZEN/CACTCGTCCACATCC/3IABkFQ/−3'
*MAP1A*-F 5'-CCTAAGTGCTGCAGACTCTT-3'
*MAP1A*-R 5'-AGGATGATGGTCTCTAGGATCTC-3'
*MAP1A*-Probe 5'-/56-FAM/TCTGGCCT/ZEN/GGGGTGTAGGA/3IABkFQ/−3'
*YEATS2*-F 5'-GAACTGACCGTAGATGAAGACA-3'
*YEATS2*-R 5'-GGAACCACAATGCTGAGAGTAG-3'
*YEATS2*-Probe 5'-TATCACAACTGCCACTTCCCCTGC-3'
*ZZZ3*-F 5'-TCTTGGACTGCTTGAGAACG-3'
*ZZZ3*-R 5'-TGAATCAGATCATGTGGCACT-3'
*ZZZ3*-Probe 5'-CAGACGATTGCTGTACTCGAGGCT-3'
*POLR2A*-F 5'-TCAGCATGTTGGACTCGATG-3'
*POLR2A*-R 5'-CGTATTCGCATCATGAACAGC-3'
*POLR2A*-Probe 5'-ACCACCTCTTCCTCCTCTTGCATCT-3'
*LAMA1*-F 5'-CTCTGGTTGCTCCTTGTGT-3'
*LAMA1*-R 5'-AGCCAGAAGTACACACATCAC-3'
*LAMA1*-Probe 5'-ATCGCCACAGTTCAGACACTTCCC-3'
*GPC4*-F 5'-CGTACTTTCGCTCAAGGCTTA-3'
*GPC4*-R 5'-TCATCTTCAACAGGGCATGG-3'
*GPC4*-Probe 5'-AGCAAGGTCTCCGTGGTAAACCC-3'
*SPP1*-F 5'-CCCCACAGTAGACACATATGATG-3'
*SPP1*-R 5'-TTCAACTCCTCGCTTTCCAT-3'
*SPP1*-Probe 5'-ACCTGACATCCAGTACCCTGATGCT-3'

### Mouse

*Ccn2*-F 5'-TTGACAGGCTTGGCGATT-3'
*Ccn2*-R 5'-GTTACCAATGACAATACCTTCTGC-3'
*Ccn2*-Probe 5'-CCGGATGCACTTTTTGCCCTTCT-3'
*Ccn1*-F 5'-GTGCAGAGGGTTGAAAAGAAC-3'

*Ccn1*-R 5'-GGAGGTGGAGTTAACGAGAAAC-3
*Ccn1*-Probe 5'-TCGGTGCCAAAGACAGGAAGCC-3'
*Fras1*-F 5'-CACTGTCACCATATCCAACGA-3'
*Fras1*-R 5'-GATGACCTTGATGCTCAGAACT-3'
*Fras1*-Probe 5'-/56-FAM/TGAAGAAGC/ZEN/CGCATACCAAGTCCG/3IABkFQ/−3'
*Errb3*-F 5'-CGTGAACTGTACCAAGATCCT-3'
*Errb3*-R 5'-GACTGTCCTGAAAACATTGAGC-3'
*Errb3*-Probe 5'-/56-FAM/ACAAGATCC/ZEN/CTGCACTGGACCC/3IABkFQ/−3'
*Scn1b*-F 5'-CATCTCCTGTAAGCGTCGTAG-3'
*Scn1b* -R 5'-GCACCTCATTCTCATAGCGTAG-3'
*Scn1b* -Probe 5'-/56-FAM/AAATTCCTC/ZEN/TGTCCCCTTCTGGCG/3IABkFQ/−3'
*Polr2a*-F 5'-GTCAGCATGTTGGACTCAATG-3'
*Polr2a*-R 5'-TCGAATCCGCATCATGAACAG-3'
*Polr2a*-Probe 5'-ACCACCTCTTCCTCCTCTTGCATCT-3'

## Luciferase reporter assay

HEK293 cells (4 × 10⁵) stably containing empty vector, *YT, YAP, YT S94A, YAP 5SA, TC, TAZ, TC S51A, or TAZ 4SA* were transfected with the pGL3b-8xGTIIC firefly luciferase reporter plasmid (containing 8 TEAD1-4 binding sites; GTIIC also known as MCAT and Hippo response element) (*Dupont et al., 2011*) and the renilla luciferase reporter plasmid (pRL-TK *Renilla*, (Promega, Madison, WI, USA)) in a well from a six-well plate (biological triplicates). Two days after transfection, the cells were collected and lysed, and extracts were assayed in technical triplicate for firefly and renilla luciferase activity using the Dual Luciferase Reporter Assay (Promega) and a Synergy H1 Hybrid Multi-Mode Microplate Reader (Biotek, Winooski, VT, USA). Each experiment was repeated at least twice.

## RNA sequencing

NIH 3T3 and SW872 cells were stably transfected, in triplicate, expressing the following proteins empty vector (EV), YAP 5SA, TFE3¸YAP-TFE3 (YT), TAZ 4SA, CAMTA1, and TAZ-CAMTA1 (TC). All cell lines, except those containing *CAMTA1*, were grown to confluence for 48 hr before collection. Cell lines containing *CAMTA1* were collected at sub-confluence. Total RNA was extracted using Trizol Reagent (Ambion-Life Technologies). Total RNA was isolated using the PureLink RNA mini kit (Invitrogen-ThermoFisher Scientific). On-column DNase (Invitrogen) treatment was performed according to manufacturer's instructions. Transcription profiling using RNA-Seq was performed by the University of Iowa Genomics Division using manufacturer recommended protocols. Initially, 500 ng of DNase I-treated total RNA was used to enrich for polyA containing transcripts using oligo(dT) primers bound to beads. The enriched RNA pool was then fragmented, converted to cDNA and ligated to sequencing adaptors containing indexes using the Illumina TruSeq stranded mRNA sample preparation kit (Cat. #RS-122–2101, Illumina, Inc, San Diego, CA). The molar concentrations of the indexed libraries were measured using the 2100 Agilent Bioanalyzer (Agilent Technologies, Santa Clara, CA) and combined equally into pools for sequencing. The concentration of the pools was measured using the Illumina Library Quantification Kit (KAPA Biosystems, Wilmington, MA) and sequenced on the Illumina HiSeq 4000 genome sequencer using 75 bp paired-end SBS chemistry.

Barcoded samples were pooled and sequenced using an Illumina HiSeq 4000 in the Iowa Institute of Human Genetics (IIHG) Genomics Core Facility. Average reads per sample for SW872 samples was 93 M (±16 M). Reads were demultiplexed and converted to FASTQ format, then processed with 'bcbio-nextgen' (https://github.com/chapmanb/bcbio-nextgen; RRID:SCR_002630 version 1.0.8). SW872 reads were aligned against the 'hg38' reference genome (genome FASTA and annotations derived from Ensembl release 78 (ftp://ftp.ensembl.org/pub/release78/fasta/homo_sapiens/dna/Homo_sapiens.GRCh38.dna.toplevel.fa.gz)). Reads from NIH 3T3 cells were aligned against 'mm10' (genome FASTA and annotations derived from ftp://ftp.ensembl.org/pub/release-97/fasta/mus_musculus/) reference genome using 'Hisat2' (*Kim et al., 2015*). Reads were also pseudo-aligned to the transcriptome using Salmon (*Patro et al., 2017*). Transcript-level abundances were converted to gene-level counts with 'tximport' (*Soneson et al., 2015*). QC was performed with 'qualimap' and 'samtools' (*Li et al., 2009*). Differential gene expression analysis was performed with DESeq2

(*Love et al., 2014*). Gene lists were analyzed using Advaita's 'iPathwayGuide' (https://www.advaita-bio.com/ipathwayguide; Advaita Bioinformatics, Ann Arbor, MI).

## ChIP sequencing

SW872 cells stably expressing Flag-tagged EV, YAP 5SA, TFE3, YT, TAZ 4SA, CAMTA1, and TC were grown to 90% confluence in 15 cm tissue culture plates (in biological triplicate). Chromatin immunoprecipitation for H3K27ac performed on SW872 cells expressing the above constructs was performed in biological duplicates. Cells were cross-linked and prepared using the SimpleChIP Enzymatic Chromatin IP Kit (Cell Signaling, #9003, Danvers, MA). Protocol was performed according to manufacturer's instructions. Lysates were sonicated for three sets of 20 s pulses in ice bath, using a VirTis Virsonic 60 Sonicator at setting 6, with a 30 s ice bath incubation between sonication pulses or Qsonic Q800R3 3 sets of 45 s pulse, 30 s rest, 50% amplitude. Chromatin digestion and concentration was verified to be within the desired range using the 2100 Agilent Bioanalyzer (Agilent Technologies, Santa Clara, CA). The SimpleChIP ChIP-seq DNA Library Prep Kit for Illumina (Cell Signaling, #56795) was used to prepare the DNA library according of manufacturer's instructions. Immunoprecipitation was performed with the anti-DYKDDDDK Tag Rabbit monoclonal antibody (Cell Signaling #14793), anti-acetyl H3K27 (Cell Signaling, clone D5E4, #81735) or provided isotype rabbit IgG control (Cell Signaling). The molar concentrations of the indexed libraries were measured using the 2100 Agilent Bioanalyzer (Agilent Technologies, Santa Clara, CA) and combined equally into pools for sequencing using an Illumina HiSeq 4000 in the Iowa Institute of Human Genetics (IIHG) Genomics Core Facility.

ChIP-seq analysis was carried out using 'bcbio-nextgen' in 'analysis: chip-seq' mode. Reads were pooled across replicates to obtain improved read depth, then trimmed for low quality and adapter read-through with 'Atropos' (version 1.1.16) (*Didion et al., 2017*) with adapter sequences ('AGATCGGAAGAG','CTCTTCCGATCT') and options set as '-quality-cutoff=5 -minimum-length=25'. Trimmed reads were aligned to the 'GRCh37/hg19' reference assembly with 'BWA-MEM' (*Li and Durbin, 2009*) using default parameters. Multi-mapping reads were removed before peak calling, and 'greylist' regions were detected and removed (https://github.com/roryk/chipseq-greylist; RRID:SCR_002630 version 1.0.8). Peaks were called using 'macs2' (*Zhang et al., 2008*) with command-line options '-f BAMPE', '-g 2.7e9', '-B', and '-q 0.10.' (FDR = 0.1) Narrow peak files were imported into the R and annotated with 'ChIPseeker' (*Yu et al., 2015*) and 'clusterProfiler' packages (*Yu et al., 2012*). Peak ranges for motif analysis were written out as FASTA using 'Biostrings' (https://bioconductor.org/packages/release/bioc/html/Biostrings.html) and 'BSgenome' packages (https://bioconductor.org/packages/release/bioc/html/BSgenome.html). Motif enrichment analysis was carried out with MEME suite (http://meme-suite.org) using the 'AME' and 'FIMO' modules and HOCOMOCO (v11) database.

## ChIP-qPCR

PCR amplification was performed in technical triplicates on the Applied Biosystems QuantStudio 3 Real-Time PCR System (Applied Biosystems-ThermoFisher). The PCR reaction contained SimpleChip Universal master mix (catalogue # 88989) Cell Signaling Technologies (Danvers MA, USA), 5 µM of primers and 2 µL of ChIP DNA or 2% input DNA. Cycling conditions as follows, 3 min at 95°C for initial denaturation followed by 40 cycles of 15 s at 95°C then 1 min at 60°C. Percent input was calculated.

The following primers were used:

*CCN2*-F 5'-GCCAATGAGCTGAATGGAGT-3'
*CCN2*-R 5'-CAATCCGGTGTGAGTTGATG-3'
*MAFK*-F 5'-GCTCTGTAAACATCGGTGACT-3'
*MAFK*-R 5'GGGAGCACCATGAGAAACTT −3'
*NEAT1*-F 5'-CTCAACAACATCCGGGAAGA −3'
*NEAT1*-R 5'-CGCTTCTCTTTAACAGTGCTTT −3'
*MAP1A*-F 5'-TGCAGAAGCAGCCAAGAA-3'
*MAP1A*-R 5'-TTTGCCGCAGTGTGTGA3'

## ATAC sequencing

A total of 50,000 cells were trypsinized and pelleted. Cells were washed once with 50 µL of cold 1x PBS buffer. After pelleting, cell pellet was resuspended in 50 µL cold lysis buffer (10 mM Tris-HCl, pH 7.4, 10 mM NaCl, 3 mM MgCl₂, 0.1% IGEPAL CA-630). The suspension was centrifuged again and supernatant discarded leaving a nuclear pellet. A transposition reaction was prepared containing 25 µL Tagment DNA buffer (#15027866), 2.5 µL TDE1 Nextera Tn5 Transposase (#15027865), and 22.5 µL Nuclease Free H₂O. The nuclear pellet was resuspended in the transposition reaction mixture. The transposition reaction was incubated at 37°C for 30 min. After transposition, fragmented DNA purified using a Qiagen miniElute PCR purification kit (catalogue #28004) per the manufacturer's instructions. Transposed DNA fragments were PCR amplified using the Nextera XT Index Kit (#15055293) per the manufacturer's instruction to generate indexed libraries.

Analysis of ATAC-seq data was carried out as follows. Reads were trimmed using 'NGmerge' in 'adapter removal' mode (*Gaspar, 2018*). Trimmed reads were then aligned as described above [command line settings: "-c 250 t 1 v 3'] and sorted with 'samtools' (*Li et al., 2009*). Peaks were called with 'genrich' (https://github.com/jsh58/Genrich; RRID:SCR_002630 version 1.0.8) in 'atacseq' mode [with flags set as '-E blacklists/wgEncodeHg19ConsensusSignalArtifactRegions.bed -v -j -r -e MT,GL000191.1,GL000192.1,GL000193.1,GL000195.1,GL000199.1,GL000205.1,GL000206.1, GL000208.1,GL000212.1,GL000214.1,GL000216.1,GL000217.1,GL000219.1,GL000220.1, GL000222.1,GL000223.1,GL000224.1,GL000225.1,GL000226.1,GL000228.1,GL000235.1, GL000243.1'.] Genrich removes PCR duplicates, excludes named regions, and blacklist regions, and properly accounts for multi-mapping reads. Peaks were called individually on each replicate. Peak files were imported along with BAM files into DiffBind (https://bioconductor.riken.jp/packages/3.0/bioc/html/DiffBind.html) (*Ross-Innes et al., 2012*) and a consensus peakset was determined. A binding affinity matrix containing (normalized) read counts for each sample at each consensus site was computed. 'DESeq2' (*Love et al., 2014*) was used to test contrasts for differential chromatin accessibility. Finally, PCA analysis was performed to show how samples cluster at differentially accessible chromatin.

## BioID mass spectrometry and SAINT analysis

Full-length YAP, TFE3, TAZ, and CAMTA1, along with YAP-TFE3 and TAZ-CAMTA1 were cloned into pcDNA5 FRT/TO FLAG-BirAR118G vectors, containing an N-terminal *E. coli* biotin ligase with an R118G mutation, as described in *Roux et al., 2012* and *Lambert et al., 2015*. These bait proteins of interest were then stably expressed in Flp-In T-REx HEK293 cells as previously described (*Couzens et al., 2013*). Expression was then induced using 1 µg/mL tetracycline for 24 hr. The cells were also treated with 50 µM biotin at the time of induction. After 24 hr, cells harvested and processed (*Lambert et al., 2015*). Two biological replicates were made for each cell line. Affinity purification and proximity biotinylation coupled to mass spectrometry were performed as described in *Lambert et al., 2015*. SAINT (significance analysis of interactome) analysis (*Choi et al., 2011*) was performed on the mass spectrometry data, using 10 controls compressed to 5. Only proteins with iProphet protein probability ≥ 0.95 were used. Recovery of bait peptides was monitored to ensure that expression levels of the bait proteins were similar (*Figure 6—figure supplement 1D,E*). Results are expressed in dotplot format. Columns show bait proteins, while the rows list the names of the identified prey proteins. Each prey protein is represented as a dot, with color signifying average spectral count, the darkness indicating average spectral count between the two biological replicates (the darker the dot, the higher the average spectral count), and the size represents the relative abundance. The ring around the dot indicates the false discovery rate (the darker the ring, the higher the confidence). The data was filtered so that each prey had a minimum of 10 spectral counts in at least one of the biological replicates. Results were also filtered to exclude components of trypsin, biotin, and streptavidin.

## Clinical samples

Six epithelioid hemangioendotheliomas, six epithelioid hemangiomas, and five epithelioid angiosarcomas were obtained from the files of the Department of Pathology, University of Iowa Hospitals and Clinics. Internal Review Board approval was obtained (IRB# 201609806).

## Statistics

For soft agar assays, statistical significance was evaluated using an unpaired two-tailed *t*-test. For poly-HEMA proliferation assays, statistical significance was evaluated using fold change increase in proliferation at day 10 with an unpaired two-tailed *t*-test. For quantitative RT-PCR, standard deviation was calculated from fold change values for each triplicate. For Kaplan-Meier curves significance was determined by Log-rank (Mantel-Cox) test. Graphs were made using the average of the technical replicates. Each experiment was repeated at least twice. For NGS studies, biological triplicates were generated by separately transducing NIH 3T3 and SW872 cells with the various contructs followed by stable selection. Hypergeometric testing was performed using the phyper() function in the stats R package (v3.6.3) set to assess enrichment and the lower tail set to false. Hypergeometric density was calculated using the related dhyper function and converted using the negative log10 of the output. For gene expression data, the population was set to the total number of recovered genes with the mean of five counts across all samples. Both the ChIP-seq and ATAC-seq, the population was set as the total number of genes annotated across all conditions. Error bars were used to define one standard deviation. For all panels, ****p<0.0001, ***p<0.001, **p<0.01, *p<0.05.

## Acknowledgements

We thank Nicholas Scalora for critical reading of the manuscript and Mariah Leidinger for performing immunohistochemistry studies. RNA-seq, ChIP-Seq, and ATAC-Seq data presented herein were obtained at the Genomics Division of the Iowa Institute of Human Genetics which is supported, in part, by the University of Iowa Carver College of Medicine. This work was supported by a University of Iowa Sarcoma Multidisciplinary Oncology Group pilot award (M.R.T), by a grant from the Veterans Health Administration Merit Review Program 1 I01 BX003644-01 (M.R.T), by a grant from the National Institutes of Health, National Cancer Institute 1 R01 CA237031-01A1 (M.R.T), and by an NCI Core Grant P30 CA086862 (University of Iowa Holden Comprehensive Cancer Center). K.F.H is a National Health and Medical Research Council Senior Research Fellow (APP1078220). A.C.G was supported by the Canadian Institutes of Health Research (FDN 144301) and the Terry Fox Research Institute and is the Tier1 Canada Research Chair in Functional Proteomics. Proteomics analysis was performed at the Network Biology Collaborative Centre at the Lunenfeld-Tanenbaum Research Institute, a facility supported by Canada Foundation for Innovation funding, by the Ontarian Government and by Genome Canada and Ontario Genomics (OGI-139).

## Additional information

### Funding

| Funder | Grant reference number | Author |
|---|---|---|
| Veterans Health Administration Merit Review Program | 1 I01 BX003644-01 | Munir R Tanas |
| National Institutes of Health | R01 CA237031-01A1 | Munir R Tanas |
| National Health and Medical Research Council | APP1078220 | Kieran F Harvey |
| Canadian Institutes of Health Research | FDN 144301 | Anne-Claude Gingras |
| University of Iowa | | Munir R Tanas |
| NCI | P30 CA086862 | Munir R Tanas |
| National Health and Medical Research Council | APP1078220 | Kieran F Harvey |
| Canadian Institutes of Health Research | FDN 144301 | Anne-Claude Gingras |
| Terry Fox Research Institute | | Anne-Claude Gingras |
| Lunenfeld-Tanenbaum Research Institute | | Anne-Claude Gingras |

| | | |
|---|---|---|
| Canada Foundation for Innovation | | Anne-Claude Gingras |
| Ontario Genomics | OGI-139 | Anne-Claude Gingras |
| National Institutes of Health | R01 CA237031-01A1S1 | Munir R Tanas |

The funders had no role in study design, data collection and interpretation, or the decision to submit the work for publication.

## Author contributions
Nicole Merritt, Investigation, Writing - original draft; Keith Garcia, Investigation, Writing - review and editing; Dushyandi Rajendran, Zhen-Yuan Lin, Xiaomeng Zhang, Katrina A Mitchell, Colleen Fullenkamp, Investigation; Nicholas Borcherding, Software, Formal analysis; Michael S Chimenti, Software, Formal analysis, Investigation, Methodology; Anne-Claude Gingras, Resources, Formal analysis, Supervision, Funding acquisition, Methodology, Writing - review and editing; Kieran F Harvey, Conceptualization, Resources, Supervision, Funding acquisition, Writing - review and editing; Munir R Tanas, Conceptualization, Resources, Formal analysis, Supervision, Funding acquisition, Investigation, Methodology, Writing - original draft, Writing - review and editing

## Author ORCIDs
Keith Garcia https://orcid.org/0000-0002-6078-5159
Anne-Claude Gingras http://orcid.org/0000-0002-6090-4437
Munir R Tanas https://orcid.org/0000-0002-6779-2642

## Ethics
Animal experimentation: This study was performed in strict accordance with the recommendations in the Guide for the Care and Use of Laboratory Animals of the National Institutes of Health. All of the animals were handled according to an approved institutional animal care and use committee (IACUC) protocol (#9052228-008 ) of the University of Iowa. All injections for mouse xenograft experiments were performed under isoflurane anesthesia, and every effort was made to minimize suffering.

## Decision letter and Author response
Decision letter https://doi.org/10.7554/eLife.62857.sa1
Author response https://doi.org/10.7554/eLife.62857.sa2

# Additional files
## Supplementary files
• Transparent reporting form

## Data availability
The accession number for the RNA-Seq data reported in this paper for NIH 3T3 cells is GEO: GSE152736. The accession number for the RNA-Seq data reported in this paper for SW872 cells is GEO: GSE152737. The accession number for the ChIP-Seq data reported in this paper is GEO: GSE152778. The accession number for the ATAC-Seq data reported in this paper is GEO: GSE152733. The accession number for the H3K27ac ChIP-Seq data reported in this paper is GEO: GSE168201. The accession number for the RNA-Seq data after YEATS2 and ZZZ3 knock-down is GEO: GSE168205.

The following datasets were generated:

| Author(s) | Year | Dataset title | Dataset URL | Database and Identifier |
|---|---|---|---|---|
| Tanas M, Chimenti M | 2020 | The TAZ-CAMTA1 and YAP-TFE3 fusion proteins modulate | https://www.ncbi.nlm.nih.gov/geo/query/acc. | NCBI Gene Expression Omnibus, GSE152736 |

| | | | | | |
|---|---|---|---|---|---|
| | | | the basal TAZ/YAP transcriptional program by recruiting the Ada2a-containing histone acetyltransferase complex [rnaseq_3T3] | cgi?acc=GSE152736 | |
| Tanas M, M | Chimenti | 2020 | The TAZ-CAMTA1 and YAP-TFE3 fusion proteins modulate the basal TAZ/YAP transcriptional program by recruiting the Ada2a-containing histone acetyltransferase complex [rnaseq_sw872] | https://www.ncbi.nlm.nih.gov/geo/query/acc.cgi?acc=GSE152737 | NCBI Gene Expression Omnibus, GSE152737 |
| Tanas M, M | Chimenti | 2020 | The TAZ-CAMTA1 and YAP-TFE3 fusion proteins modulate the basal TAZ/YAP transcriptional program by recruiting the Ada2a-containing histone acetyltransferase complex [chipseq] | https://www.ncbi.nlm.nih.gov/geo/query/acc.cgi?acc=GSE152778 | NCBI Gene Expression Omnibus, GSE152778 |
| Tanas M, M | Chimenti | 2020 | The TAZ-CAMTA1 and YAP-TFE3 fusion proteins modulate the basal TAZ/YAP transcriptional program by recruiting the Ada2a-containing histone acetyltransferase complex [atacseq] | https://www.ncbi.nlm.nih.gov/geo/query/acc.cgi?acc=GSE152733 | NCBI Gene Expression Omnibus, GSE152733 |
| Tanas M, M | Chimenti | 2020 | The TAZ-CAMTA1 and YAP-TFE3 fusion proteins modulate the basal TAZ/YAP transcriptional program by recruiting the Ada2a-containing histone acetyltransferase complex [ChIP-seq] | https://www.ncbi.nlm.nih.gov/geo/query/acc.cgi?acc=GSE168201 | NCBI Gene Expression Omnibus, GSE168201 |
| Tanas M, M | Chimenti | 2020 | The TAZ-CAMTA1 and YAP-TFE3 fusion proteins modulate the basal TAZ/YAP transcriptional program by recruiting the Ada2a-containing histone acetyltransferase complex [RNA-seq] | https://www.ncbi.nlm.nih.gov/geo/query/acc.cgi?acc=GSE168205 | NCBI Gene Expression Omnibus, GSE168205 |

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
