## [Decision Letter]

**Acceptance summary:**

This manuscript characterizes the molecular functions of two fusion proteins, YAP-TFE3 and TAZ-CAMTA1, that contribute to the vascular sarcoma epithelioid hemangioendothelioma (EHE). The manuscript demonstrates that expression of these fusion proteins promote pro-tumorigenic properties in various cell types, including increased proliferation and anchorage independent growth, as well as increased growth in subcutaneous xenograft experiments in mice. Novel aspects of the study include the analysis of YAP-TFE3, which has not yet been examined on a molecular level, mapping the genomic landscape of TAZ-CAMTA1 and YAP-TFE3, and identifying new associated chromatin modulators that distinctly contribute to pro-tumorigenic functions of the fusion proteins. The study is well conducted and identifies new mechanisms by which the TAZ-CAMTA1 and YAP-TFE3 fusion proteins contribute to the development of EHE.

**Decision letter after peer review:**

Thank you for submitting your article "TAZ-CAMTA1 and YAP-TFE3 alter the TAZ/YAP transcriptome by recruiting the ATAC histone acetyltransferase complex" for consideration by *eLife*. Your article has been reviewed by 2 peer reviewers, and the evaluation has been overseen by a Reviewing Editor and Kevin Struhl as the Senior Editor.

The reviewers have discussed the reviews with one another and the Reviewing Editor has drafted this decision to help you prepare a revised submission.

Summary:

The manuscript by Merritt and Garcia et al. characterizes the molecular functions of two fusion proteins, YAP-TFE3 and TAZ-CAMTA1, that contribute to the vascular sarcoma Epithelioid hemangioendothelioma (EHE). The authors show that ectopic expression of the fusion proteins in NIH3T3 fibroblasts and SW872 liposarcoma cells promote pro-tumorigenic properties, such as increased proliferation, anchorage independent growth, and increased growth in subcutaneous xenograft experiments in mice. Novel aspects of the study include the analysis of YAP-TFE3, which has not yet been examined on a molecular level, mapping the genomic landscape of TAZ-CAMTA1 and YAP-TFE3, and identifying new associated chromatin modulators that distinctly contribute to pro-tumorigenic functions of the fusion proteins. The study is generally well conducted and identifies potential new mechanisms by which the TAZ-CAMTA1 and YAP-TFE3 fusion proteins contribute to the development of EHE.

Overall, the Reviewers agreed to recommend submission of a revised manuscript addressing the following points:

Essential revisions:

1. Given the comparisons between the various fusion and non-fusion proteins throughout many experiments, it is important to show similar expression levels of the respective proteins across the cell lines that were generated. Figure S1A and S1B show expression, but based on what is shown it is not possible to assess the similarities in expression between the cell lines. If the authors want to make arguments that post-transcriptional mechanisms may account for differences, they should show data to support this.

2. It is difficult to draw strong conclusions from the nuclear-cytoplasmic fractionation of YAP and YAP-TFE3 in FiguresS2C and S2D since these data were not compared side-by-side. Further, the effects of confluence on the nuclear abundance of YAP vs YAP-TFE3 in Figure S2C is not very convincing. Please address.

3. The potential enrichment of EGR2 for TAZ-CAMTA1 that is suggested by the ChIP-seq analysis is interesting and should be biochemically tested. Further, the authors indicate that the YAP-TFE3 ChIP-seq analyses indicates enrichment for the MiTF motif, which suggests that the TFE3 domain mediates DNA binding. Prior studies have shown that the bHLH and LZ domains in TFE3 mediate homo and hetero association with MiTF family proteins to mediate DNA-binding. The authors should test whether YAP-TFE3 associates with TFE3 or other MiTF family members and/or test whether these domains are required for YAP-TFE3 oncogenic function.

4. The authors should better describe what is uniquely enriched in the YAP-TFE3 and TAZ-CAMTA1 gene expression programs and define whether these unique changes rely on ATAC complex component association. For example, are these genes altered in expression following YEATS2 or ZZZ3 knockdown?

5. The identification of YEATS2 and ZZZ3 as potential fusion protein associated chromatin remodeling factors is very interesting given the known functions of the ATAC complex. The authors should directly test whether YAP-TFE3 and/or TAZ-CAMTA1 modify ATAC complex activity (e.g. H3K9 acetylation on a subset of direct targets that they identified), comparing any findings to TAZ-4SA or YAP-5SA.

6. While it is interesting that YEATS2 and ZZZ3 expression is associated with poor prognosis across sarcomas, this does not necessarily link to YAP/TAZ or fusion protein activity in EHE. The study would greatly benefit from directly linking some of the discoveries made in the manuscript to the in vivo biology of EHE. For example, are any of the identified fusion protein gene targets (ideally direct targets that are uniquely expressed by the fusion proteins) elevated in EHE tissue biopsies?

7. Figure 1C-D, 1H. The YT and TC fusions regulate cell proliferation in vitro. However, this is inconsistent with in vivo data in which the fusions only seem to impact the rate of tumor initiation while the rate of tumor growth was similar between EV and the YT or TC fusions. Figure S3 also shows that Ki67 staining is the same between all three groups (actually lower in YT). Please address.

8. Figure 1J. The EV control in the SW872 model seems to not be tumorigenic. Others have used this cell line for xenograft studies (Stratford et al. Sarcoma. 2012; Zhang et al. Cancer Res. 2013; Li et al. Transl Oncol. 2014). Could there have been problems with the EV that impacted cell viability? Would EV tumors have eventually emerged if the experiment was carried out longer? Please address.

9. Figure 2. The authors reference their previous work showing the oncogenic properties of the TC fusion is dependent on TEAD4 but the data reported here focuses on TEAD1. Is this due to tissue specificity or a general dependency on both TEAD1/4? Last, what is the dominant band in the YAP-TFE3 S94A immunoblot in Figure 2D?

10. Figure 3-5. Ideally ChIP-seq would be performed on endogenous proteins as overexpression can induce non-specific and altered DNA binding profiles (Kidder et al. Nat Immunol. 2011.) However, the lack of EHE cell lines precludes using endogenous proteins. In this case, it would be helpful to combine the overexpression vectors with knockdown of TEADs or ATAC members to show that the described DNA binding and chromatin accessibility are specific.

11. The authors commented that the genomic occupancy of TC and YT fusions is predominantly at enhancers. What histone markers were used to identify enhancer regions?

12. Figure 6C. The profile of chromatin modifiers for wildtype YAP and TAZ is minimal compared to the fusions. How was the relative expression of proteins normalized for BioID? Could the differences in interactors be explained by massive overexpression of the fusions in comparison to WT?

13. Figure 7-8. Does knockdown of YEATS2 or ZZZ3 alter expression of canonical TEAD-dependent YAP/TAZ target genes such as CTGF or CYR61? Based on the ATAC-Seq data, is there a subset of genes that one could probe to determine if loss of YEATS2 or ZZZ3 impacts their expression? The co-IPs should be performed using wildtype YAP and TAZ as negative controls to confirm that interactions with the ATAC complex occur via contributions from CAMTA1 and TFE3.

---

## [Author Response]

Essential revisions:1. Given the comparisons between the various fusion and non-fusion proteins throughout many experiments, it is important to show similar expression levels of the respective proteins across the cell lines that were generated. Figure S1A and S1B show expression, but based on what is shown it is not possible to assess the similarities in expression between the cell lines. If the authors want to make arguments that post-transcriptional mechanisms may account for differences, they should show data to support this.Thank you, we agree with the above comments. We have repeated the above mentioned experiments in Figure S1 showing expression of the various constructs on the same gel including a loading control (β-actin). As noted previously, we do see variability in expression of the various proteins, but that the phenotypes we have observed run counter to differences in expression (e.g. YAP-TFE3 drives greater anchorage independent growth than YAP-TFE3 S94A despite YAP-TFE3 S94A demonstrating higher expression). We have in the past attempted modulating multiplicity of infection when transducing cell lines with the various constructs, and have been unable to standardize expression levels, suggesting that the differences in protein stability may be accounting for those differences in expression.We have also made sure to test various hypothesis from multiple angles/approaches supporting the conclusions that have been made (e.g. the importance of TEAD binding to YAP-TFE3 was demonstrated by (a) TEAD knock-down experiments and anchorage independent growth experiments, (b) qRT-PCR experiments with the YAP-TFE3 S94A mutant, (c) ChIP-qPCR with the YAP-TFE3 S94A mutant, and (d) luciferase reporter assay with the YAP-TFE3 S94A mutant.)2. It is difficult to draw strong conclusions from the nuclear-cytoplasmic fractionation of YAP and YAP-TFE3 in FiguresS2C and S2D since these data were not compared side-by-side. Further, the effects of confluence on the nuclear abundance of YAP vs YAP-TFE3 in Figure S2C is not very convincing. Please address.We agree this could have been more clearly demonstrated. Nuclear fractions for YAP and YAP-TFE3 were run on the same gel and cytoplasmic fractions for YAP and YAP-TFE3 were run together on a separate gel. They are separated in Figures S2C and S2D to facilitate comparison of nuclear levels of YAP during sparse vs. confluent plating, and nuclear levels of YAP-TFE3 during sparse vs. confluent plating (YAP-TFE3 requires a longer exposure than YAP since YAP-TFE3 is expressed at lower levels). This was performed to illustrate how YAP-TFE3 remains within the nucleus under confluent conditions (because it is not being negatively regulated by the Hippo pathway) while YAP expression in the nucleus decreases (because it is regulated normally by the Hippo pathway). We agree that the previous blot was overexposed, and so we have reprobed the membrane and are including a more appropriately exposed blot in Figures S2C and S2D.3. The potential enrichment of EGR2 for TAZ-CAMTA1 that is suggested by the ChIP-seq analysis is interesting and should be biochemically tested. Further, the authors indicate that the YAP-TFE3 ChIP-seq analyses indicates enrichment for the MiTF motif, which suggests that the TFE3 domain mediates DNA binding. Prior studies have shown that the bHLH and LZ domains in TFE3 mediate homo and hetero association with MiTF family proteins to mediate DNA-binding. The authors should test whether YAP-TFE3 associates with TFE3 or other MiTF family members and/or test whether these domains are required for YAP-TFE3 oncogenic function.Thank you for these comments. To address the point raised regarding the interaction of MiTF family members, we noted that by our BioID mass spectrometry results (Figure S9) that TFE3 was the chief MiTF family candidate that could potentially interact with YAP-TFE3 directly. To test this hypothesis, we immunoprecipitated TFE3 and probed for YAP-TFE3 and showed that YAP-TFE3 does interact with TFE3 (see Figure S6A and discussed in lines 281-285). To determine whether the bHLH/leucine zipper domain is required for YAP-TFE3’s oncogenic function, we evaluated the function of the YAP-TFE3 R358K-I354N (disrupts helix 1 of the bHLH domain) and showed that it altered anchorage independent growth (Figure S6A and discussed in lines 281-285). Additional studies are warranted to elucidate the functional role of the bHLH domain further, and anticipate doing so in a subsequent manuscript.To address the potential enrichment of EGR2 for TAZ-CAMTA1 suggested by the ChIP-seq analysis, we performed co-immunoprecipitation studies for TAZ-CAMTA1 and EGR2. By co-immunoprecipitation, we did not observe an interaction between TAZ-CAMTA1 and EGR2, suggesting that TAZ-CAMTA1 and EGR2 do not directly interact with one another. It is possible another protein may bridge the interaction between TAZ-CAMTA1 and EGR2, which is something we can follow up on in future studies.4. The authors should better describe what is uniquely enriched in the YAP-TFE3 and TAZ-CAMTA1 gene expression programs and define whether these unique changes rely on ATAC complex component association. For example, are these genes altered in expression following YEATS2 or ZZZ3 knockdown?We agree this is an important point. To address this, we performed RNA-Seq on SW872 TAZ-CAMTA1 and YAP-TFE3 expressing cells with either ZZZ3 or YEATS2 knock-down and compared the resultant transcriptomes to non-targeting controls. We found by principal component analysis (Figure 8A) that both knock-down of YEATS2 or ZZZ3 altered the transcriptome of both TAZ-CAMTA1 and YAP-TFE3. Comparison of genes within the TAZ-CAMTA1 and YAP-TFE3 transcriptomes with the largest decrease in expression after knock-down of YEATS2 and ZZZ3 showed that many of them were extracellular matrix proteins (Figure 8B) and were validated by qRT-PCR (Figure 8C); discussed in lines 416-419. Additional pathway analysis on the above RNA-Seq experiment showed that pathways (e.g. MAPK and PI3K pathways) activated by YAP-TFE3 (Figure 3I) are inactivated with YEATS2 and ZZZ3 knock-down (Figure 8D and Figure S12A-D); discussed in lines 419-424.5. The identification of YEATS2 and ZZZ3 as potential fusion protein associated chromatin remodeling factors is very interesting given the known functions of the ATAC complex. The authors should directly test whether YAP-TFE3 and/or TAZ-CAMTA1 modify ATAC complex activity (e.g. H3K9 acetylation on a subset of direct targets that they identified), comparing any findings to TAZ-4SA or YAP-5SA.We agree that the proposed studies are important, but would be most informative if performed at a genome-wide level (e.g. H3K9ac ChIP-seq) and in conjunction with other mechanistic studies dissecting the function of the ATAC complex, which we anticipate doing in our follow-up studies to this manuscript. After additional editorial correspondence, we have instead elected to focus on dissecting the function of the ATAC complex in a related experiment for the current manuscript, the RNA-Seq experiment delineated above in response to point 4, since it would more directly address the main thesis of the manuscript which is focused on the transcriptomic changes the ATAC complex promotes.6. While it is interesting that YEATS2 and ZZZ3 expression is associated with poor prognosis across sarcomas, this does not necessarily link to YAP/TAZ or fusion protein activity in EHE. The study would greatly benefit from directly linking some of the discoveries made in the manuscript to the in vivo biology of EHE. For example, are any of the identified fusion protein gene targets (ideally direct targets that are uniquely expressed by the fusion proteins) elevated in EHE tissue biopsies?Thank you for this suggestion. To address this, we have performed immunohistochemistry for various extracellular matrix proteins, enriched in pathway analysis performed on the TAZ-CAMTA1 and YAP-TFE3 transcriptional programs. We performed immunohistochemistry for COL1A1 (Collagen Type I Alpha 1 Chain), CTGF (Connective Tissue Growth Factor), FBLN5 (Fibulin-5), and VTN (vitronectin) on 6 epithelioid hemangioendothelioma (EHE) FFPE samples as well as two histological mimics: epithelioid angiosarcoma (E-AS) and epithelioid hemangioma (EH). We showed that COL1A1, CTGF, FBLN5, and VTN were widely expressed in all of the EHE samples evaluated (Figures8H and Figure S12E-F). In addition, VTN appears to be specifically upregulated in EHE in a statistically significant manner as compared to E-AS and EH controls (Figure 8H).7. Figure 1C-D, 1H. The YT and TC fusions regulate cell proliferation in vitro. However, this is inconsistent with in vivo data in which the fusions only seem to impact the rate of tumor initiation while the rate of tumor growth was similar between EV and the YT or TC fusions. Figure S3 also shows that Ki67 staining is the same between all three groups (actually lower in YT). Please address.We agree with the above points that have been raised. TAZ-CAMTA1 and YAP-TFE3 promote proliferation in the very specific context of anchorage independent growth which captures only some aspects of tumor growth in vivo. In vitro experiments in cancer cell lines can give mechanistic insight, but cannot recapitulate every aspect of in vivo biology (Liu K, et al. Nat Commun 2019; 10 (2138); Gould S, et al. Nat Med 2015; 21:431-439). This is addressed in the revised manuscript in lines 147-149.8. Figure 1J. The EV control in the SW872 model seems to not be tumorigenic. Others have used this cell line for xenograft studies (Stratford et al. Sarcoma. 2012; Zhang et al. Cancer Res. 2013; Li et al. Transl Oncol. 2014). Could there have been problems with the EV that impacted cell viability? Would EV tumors have eventually emerged if the experiment was carried out longer? Please address.

The above is entirely correct. We have carried the experiment out longer, and empty vector tumors will eventually emerge. To address this, a discussion of the above has been included in the revised manuscript in lines 155-157, which reads: *“*In the mildly tumorigenic SW872 cell line (above references added), expression of YAP-TFE3 and TAZ-CAMTA1 decreased latency to tumor formation compared to empty vector in vivo (**Figure 1J**), similar to NIH 3T3 cells.”

9. Figure 2. The authors reference their previous work showing the oncogenic properties of the TC fusion is dependent on TEAD4 but the data reported here focuses on TEAD1. Is this due to tissue specificity or a general dependency on both TEAD1/4? Last, what is the dominant band in the YAP-TFE3 S94A immunoblot in Figure 2D?The mass spectrometry dot plot (Figure S9) showed that TAZ-CAMTA1 interacted with TEAD4, but that YAP-TFE3 interacted with predominantly TEAD1 and TEAD3 but not TEAD4; hence leading us to evaluate and confirm the interaction of YAP-TFE3 with TEAD1 by co-IP in Figure 2. The BioID mass spectrometry experiment for YAP-TFE3 and TAZ-CAMTA1 was performed in the same cell line (HEK293 cells) which argues against the interaction with different TEAD transcription factors being due to tissue specificity, and suggests this is due to intrinsically different protein binding properties of the fusion proteins themselves. We anticipate that the dominant band in the YAP-TFE3 S94A blot is an unstable degradation product. The above has been addressed in the revised manuscript in lines 180-182.10. Figure 3-5. Ideally ChIP-seq would be performed on endogenous proteins as overexpression can induce non-specific and altered DNA binding profiles (Kidder et al. Nat Immunol. 2011.) However, the lack of EHE cell lines precludes using endogenous proteins. In this case, it would be helpful to combine the overexpression vectors with knockdown of TEADs or ATAC members to show that the described DNA binding and chromatin accessibility are specific.We have previously addressed the specificity of binding the TEAD consensus binding sequence at least one way with the luciferase reporter assay in Figure 2I. To further address the specificity of TEAD mediated DNA binding, we have performed ChIP-qPCR for CTGF (a well-established TEAD-bound gene) in SW872 cells overexpressing YAP-TFE3 or YAP-TFE3 S94A (mutant that abrogates TEAD binding) shown in Figure 2H. ChIP-qPCR for the CTGF promoter (contains the TEAD binding consensus sequence) showed a 6-fold decrease in CTGF promoter binding by YAP-TFE3 S94A overexpressing cells as compared to YAP-TFE3 control, discussed in lines 185-187. The above study furthers shows that the TEAD binding profile described for YAP-TFE3 is not an artifact of overexpression but rather a property of the fusion protein itself.Evaluating chromatin accessibility by ATAC-Seq as a function of ZZZ3 or YEATS2 knock-down is important to do and will be part of future studies. For this current manuscript, we have chosen to address the specificity of chromatin accessibility in a way that is in keeping with the main message of the manuscript which is focused on how the ATAC complex modulates the TAZ-CAMTA1 and YAP-TFE3 transcriptomes. To do this we have performed the RNA-Seq experiment mentioned in point 4 and identified the resultant transcriptomes after YEATS2 and ZZZ3 knock-down. We have then intersected differentially expressed genes (DEGs) after YEATS2 and ZZZ3 knock-down in YAP-TFE3 expressing cells and intersected them with transposase accessible areas of the genome identified by ATAC-Seq (Figure 8E and 8F). We showed that 71% of DEGs after YEATS2 knock-down were present within transposase accessible areas of the genome (Figure 8E), while 74% of DEGs after ZZZ3 knock-down were present within transposase accessible portions of the genome (Figure 8F), linking the described chromatin accessibility due to expression of YAP-TFE3 to ATAC-dependent components of the YAP-TFE3 transcriptome. This is discussed in lines 424-428 of the revised manuscript.11. The authors commented that the genomic occupancy of TC and YT fusions is predominantly at enhancers. What histone markers were used to identify enhancer regions?Distal intergenic sequences, which are known to contain sequences functioning as enhancers were defined as sequences greater than 3kb (Zhang Y, et al. Nature 2013; 504(7479): 306-310) from the assigned gene closest to them. To further define the location of enhancer regions, we performed ChIP-seq for H3K27ac (Calo E, et al. Mol Cell 2013; 49(5): 825-837; Creyghton MP, et al. Proceedings of the National Academy of Sciences of the United States of America 2010; 107: 21931-21936) in SW872 cells expressing the fusion proteins and controls, and overlapped with the chromatin binding profile of TAZ-CAMTA1, YAP-TFE3, and controls. By overlapping H3K27ac ChIP peaks with ChIP peaks for the various constructs, we confirmed that distal intergenic sequences were predominantly composed of active enhancer sequences (69% overlap with TAZ-CAMTA1 distal intergenic sequences and 68% overlap in YAP-TFE3 distal intergenic sequences) (Figure 4D and E). This discussion including the above references have been included in lines 248-261 of the revised manuscript.12. Figure 6C. The profile of chromatin modifiers for wildtype YAP and TAZ is minimal compared to the fusions. How was the relative expression of proteins normalized for BioID? Could the differences in interactors be explained by massive overexpression of the fusions in comparison to WT?We agree this is an important point. While we do not systematically normalize the BioID data to account for differential protein expression (which could introduce unwanted bias), we do monitor the recovery of the bait peptides to ensure that we do not have massive differences in expression levels. Here, for YAP and YAP-TFE3, analysis of spectra for shared peptides showed no clear difference in relative abundance, while there was a ~2-fold increase in peptide detection for TAZ-CAMTA1 as compared to TAZ. While it is not impossible that these changes in relative expression increase some of the associations, this is not likely to explain the large changes in the recruitment of chromatin modifiers detected in the study. We have added the protein quantification tables and associated method text in the revised manuscript (Figure S9 D,E and lines 937-938).13. Figure 7-8. Does knockdown of YEATS2 or ZZZ3 alter expression of canonical TEAD-dependent YAP/TAZ target genes such as CTGF or CYR61? Based on the ATAC-Seq data, is there a subset of genes that one could probe to determine if loss of YEATS2 or ZZZ3 impacts their expression? The co-IPs should be performed using wildtype YAP and TAZ as negative controls to confirm that interactions with the ATAC complex occur via contributions from CAMTA1 and TFE3.As mentioned above, to address whether knock-down of YEATS2 or ZZZ3 affected expression of CTGF or CYR61, we performed RNA-Seq on TAZ-CAMTA1 or YAP-TFE3 expressing cells with either YEATS2 or ZZZ3 knock-down. Of the different combinations, only YAP-TFE3 expressing cells with YEATS2 or ZZZ3 knock-down demonstrated a modest reduction of CTGF (log_2_ of -0.64 and -0.91, respectively) (Table S7). As shown in Figures 8B and 8D genes encoding extracellular matrix proteins or resulting in activation of PI3K-Akt signaling were affected by ZZZ3 or YEATS2 knock-down. As shown in Figure 8E 71% of DEGs after YEATS2 knock-down were present within transposase accessible areas of the genome as defined by ATAC-Seq. Similarly, 74% of DEGs after ZZZ3 knock-down were present within transposase accessible portions of the genome. The above discussion has been included in lines 410-428. We have confirmed that TAZ-CAMTA1 co-immunoprecipitates with the KAT14 subunit of the ATAC complex while the TAZ control does not (Figure S 11E and lines 407-409).